# Distinct nociception processing in the dysgranular and barrel regions of the mouse somatosensory cortex

Hironobu Osaki [1,2 ✉], Moeko Kanaya[1], Yoshifumi Ueta [1] & Mariko Miyata [1 ✉]

Nociception, a somatic discriminative aspect of pain, is, like touch, represented in the primary somatosensory cortex (S1), but the separation and interaction of the two modalities within S1 remain unclear. Here, we show spatially distinct tactile and nociceptive processing in the granular barrel field (BF) and adjacent dysgranular region (Dys) in mouse S1. Simultaneous recordings of the multiunit activity across subregions revealed that Dys neurons are more responsive to noxious input, whereas BF neurons prefer tactile input. At the single neuron level, nociceptive information is represented separately from the tactile information in Dys layer 2/3. In contrast, both modalities seem to converge on individual layer 5 neurons of each region, but to a different extent. Overall, these findings show layer-specific processing of nociceptive and tactile information between Dys and BF. We further demonstrated that Dys activity, but not BF activity, is critically involved in pain-like behavior. These findings provide new insights into the role of pain processing in S1.

[1] Division of Neurophysiology, Department of Physiology, Graduate School of Medicine, Tokyo Women's Medical University, Shinjuku, Tokyo, Japan. [2] Present address: Laboratory of Functional Brain Circuit Construction, Graduate School of Brain Science, Doshisha University, Kyotanabe, Kyoto, Japan. ✉email: hosaki@mail.doshisha.ac.jp; mmiyata@twmu.ac.jp

The primary somatosensory cortex (S1) plays a central role in tactile information processing[1]. The tactile representation in S1 is orderly arranged in a somatotopic fashion[2]. On the other hand, S1 is responsible for somatic discriminative aspects of pain processing, such as the location, intensity, and quality of pain[3–10]. S1 receives thalamocortical nociceptive information[11] and relays it to other pain-related cortical areas, such as the anterior cingulate cortex, which is responsible for the affective aspects of pain[12,13]. S1 also modulates noxious inputs via the corticotrigeminal[14] and corticospinal[15] pathways under both acute and chronic pain conditions. Therefore, S1 can be viewed as a network hub of pain processing and a target for interventions to control pain. However, it remains unclear how S1 processes nociceptive information and somatic tactile information distinctively.

Mouse S1 is divided into two subregions based on its cytoarchitecture: the granular region known as the barrel field (BF), which is identified by unique clusters of layer (L4) neurons, and the adjacent dysgranular region (Dys), which has poorly defined L4[6,16]. The two subregions are thought to be functionally different. For instance, BF is the center for processing tactile input from whiskers[17–19], while Dys receives proprioceptive input from deep muscle stimulations or joint rotations[20,21]. In nociception, BF neurons in deeper layers receive noxious inputs[11,22,23]. Similarly, Dys neurons in deeper layers respond to noxious pinching and pruriceptive inputs[24,25]. However, how each subregion processes nociceptive information with/without tactile information remains unknown because a direct comparison between the two subregions is lacking.

Here, we found that nociceptive and tactile information tends to be separately represented in Dys and BF, respectively, by simultaneously recording both subregions. Dys was also predominantly activated under neuropathic pain conditions produced by peripheral nerve injury. Reflecting the spatially distinct representation of nociception, the optogenetic inhibition of neuronal activity of Dys, but not of BF, reduced pain-like behavior induced by noxious inputs. Thus, we clarified a distinct functional role in pain processing in Dys, which generates proper escape behavior from noxious inputs.

## Results

**Nociceptive information processing in Dys and BF.** First, we compared the response properties between Dys and BF during noxious heat stimuli (noxH; 45–50°C) applied to a whisker pad (Fig. 1a). Although the septa within BF belong to the dysgranular region in terms of the cytoarchitecture, we designated the dysgranular zone surrounding BF as Dys. We recorded multiunit activities (MUA) simultaneously from Dys and BF neurons in L2/3, L4, L5a and L5b. The MUA in Dys increased in all recorded layers when the temperature of the Peltier device reached a noxious range (45–50 °C), while the responses differed among layers in BF: MUA to noxH did not increase in L2/3 or L4 but increased slightly in L5a (Fig. 1b). To evaluate the preference to noxH, the signal-to-noise ratio (S/N; see Methods) was calculated. The response to noxH (labeled S, beige-shaded region in Fig. 1b) was used as the signal, and the response to an innoxious heat range (33–45 °C, labeled N, grey-shaded region in Fig. 1b) was used as the noise. Comparisons of simultaneously recorded neural pairs showed that Dys neurons are more responsive to noxH than BF neurons (Fig. 1c, $P = 0.0056$ for L2/3, 0.0056 for L4, and 0.049 for L5a, $n = 8$ animals). When comparing S/N values for the same layers between Dys and BF, the S/N in Dys was significantly higher than in BF in L2/3 ($P = 0.89 \times 10^{-4}$, multiple-comparisons test, Fig. 1d). Within BF, the S/N was significantly higher in L5a than in L2/3 ($P = 0.03$, Fig. 1d). This difference between layers for noxH responses in BF is

consistent with previous studies[22,23,26]. This trend was also observed when the response of the Peltier device at steady-state temperature (around 30 °C) was used as the noise signal to calculate the S/N. Together, the MUA of Dys showed greater sensitivity to noxH than BF (Fig. 1e). We also examined the expression of c-Fos, a marker for neuronal activity, in the area within S1 that responds to noxious input following the injection of capsaicin into the whisker pad (Supplementary Fig. 1a). To differentiate between Dys and BF, we co-immunostained with NeuN and VGluT2, markers for neurons and thalamocortical terminals, respectively (Supplementary Fig. 1b). The number of c-Fos-positive neurons increased significantly in L4 of Dys ($P = 0.0027$) and L5a of BF ($P = 0.0249$, Supplementary Fig. 1c). Thus, we confirmed noxious inputs activated superficial layers in Dys and L5a in BF.

We next assessed the response preference to tactile stimuli by comparing the S/N in response to whisker deflections (see Methods). Neurons in BF responded precisely to the onset of each whisker deflection, whereas those in Dys did not (Fig. 2a, b). The S/N to whisker deflections was higher in BF than Dys in each layer (Fig. 2c, d), indicating that BF responses were stronger than Dys responses to tactile stimuli (Fig. 2e). Since the whisker pad was directly stimulated by heat, we tested its response to tactile stimulation by placing a paint brush into the whisker pad (this is the equivalent position as the heat stimulus) and confirmed that BF prefers tactile stimuli to Dys (Supplementary Fig. 2).

**Nociceptive information is processed separately from tactile information between Dys and BF L2/3.** The results of the MUA analysis indicated that there are some spatial biases for nociceptive information processing between Dys and BF (Fig. 1). Thus, we examined the modality specificity of single neurons in two subregions. L2/3 and 5 are the cortical output layers that make connections with the motor and anterior cingulate cortices, which are implicated in pain-like behavior[13,27]. Peristimulus time histograms (PSTHs) of well-isolated neurons in L2/3 simultaneously recorded from the two subregions are shown in Fig. 3a; recorded neurons were sorted by the time of the peak response to a heat stimulus. In Dys L2/3, the steepness of the peak responses to a heat stimulus increased within the noxious heat range, indicating that the majority of Dys neurons responded to the noxious heat. By contrast, only a few neurons responded in BF L2/3 (white box in Fig. 3a, left column). On the other hand, many BF L2/3 neurons responded well to tactile stimuli. Notably, noxious heat-responding neurons in Dys L2/3 did not respond to tactile stimuli (white box in Fig. 3a, right column).

The steepness of the peak responses to heat stimuli increased within the noxious heat range in both regions in deeper layers (white boxes in L5a and L5b in Fig. 3a, left column). This tendency indicates that nociceptive neurons were increased in deeper layers. Except for Dys L5a, nociceptive neurons in L5 tended to respond also to tactile stimuli (white boxes in Fig. 3a, right column). Thus, Dys neurons in L2/3 process nociceptive information separately from tactile information, whereas neurons in L5 tend to respond to both tactile and noxious inputs.

To quantify these observations, we classified the neurons into nociceptive, tactile, integrative, and non-reactive types according to the S/N to noxH and tactile stimuli (Fig. 3b and Supplementary Fig. 3). The distributions of the S/N clustered according to the median S/N for all recorded neurons (1.46 for nociceptive and 1.81 for tactile, Supplementary Fig. 3b, d, f, g). In addition, normalized PSTHs of neurons classified according to these values represented the characteristics of each neuron group (Fig. 3b): nociceptive-type neurons responded to noxious heat but not to tactile stimuli, tactile-type neurons responded to tactile inputs but not to noxious heat, and integrative-type neurons responded to both stimuli.

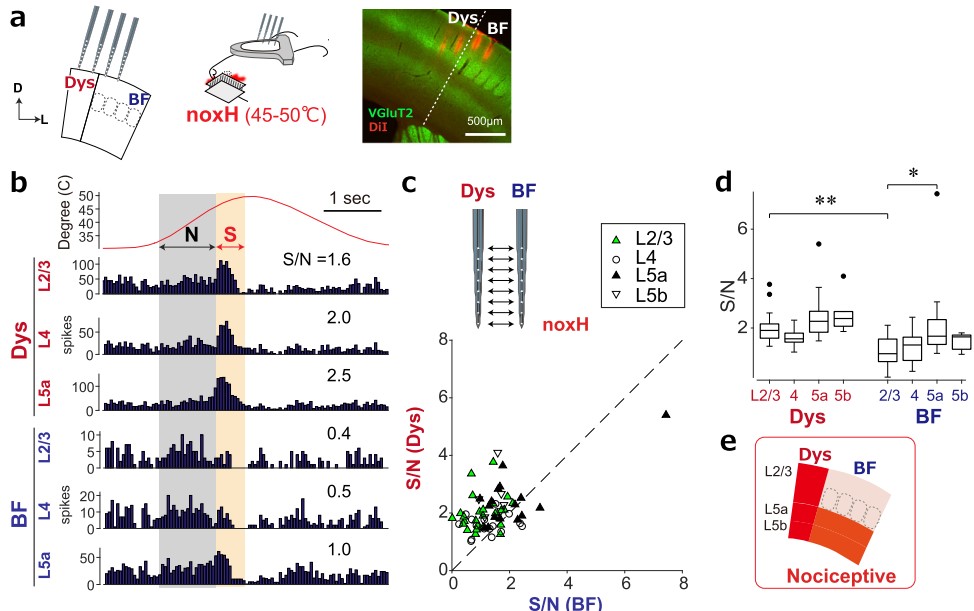

**Fig. 1 Dysgranular region in S1 responds to noxious input. a** Setup for simultaneous recordings of Dys and BF while applying noxious heat stimuli (noxH) to the left whisker pad. A brain section showing electrode tracks (DiI, red). VGluT2 (green) staining shows the border of Dys and BF. Scale bar, 500 μm. **b** PSTHs of representative MUA recordings to noxH in L2/3, 4, and 5a of Dys and BF. The shaded areas indicate regions for calculating the S/N ratios. **c** A scatter plot of S/N of simultaneously recorded multiunit activities in response to a noxH stimulus. $P = 0.000023$ for L2/3 ($n = 25$), 0.0056 for L4 ($n = 17$), 0.049 for L5a ($n = 16$) and 0.69 for L5b ($n = 6$) by the two-sided Wilcoxon signed-rank test. The diagonal indicates unity. **d** Statistical comparison of S/N responses to noxH. *$P = 0.030$, **$P = 0.89 \times 10^{-4}$ by the Kruskal–Wallis test followed by Sidak's test; $n = 8$ animals. Box plots (median with the 25th and 75th percentiles). Data points beyond the whiskers are shown by the dots. **e** A summary diagram indicating nociceptive regions in S1.

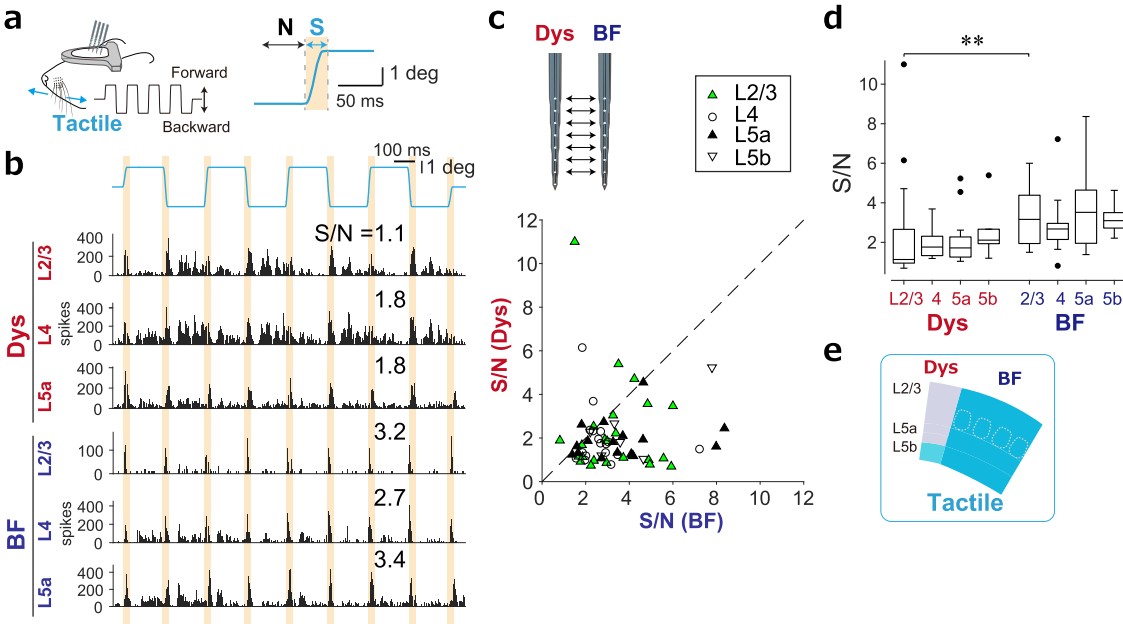

**Fig. 2 BF responds more selectively to tactile input than Dys. a** Setup for recording responses of Dys and BF to tactile stimulation of the whiskers. The shaded area indicates the time used as the signal (S) and noise (N) to calculate the S/N of the tactile input. **b** Examples of PSTHs of MUA to tactile stimuli in L2/3, 4, and 5a of Dys and BF recorded at the same time (see also Fig. 1e). **c** A scatter plot of S/N of simultaneously recorded MUA to tactile stimuli. $P = 0.0038$ for L2/3 ($n = 25$), 0.028 for L4 ($n = 17$), 0.0002 for L5a ($n = 16$), and 0.063 for L5b ($n = 6$) by the two-sided Wilcoxon signed-rank test. The diagonal indicates unity. **d** S/N to a tactile stimulus was higher in BF L2/3. **$P = 5.4 \times 10^{-4}$ by the Kruskal–Wallis test followed by Sidak's test; $n = 8$ animals. Box plots (median with the 25th and 75th percentiles). Data points beyond the whiskers are shown by the dots. **e** A summary diagram indicateing tactile preference regions in S1.

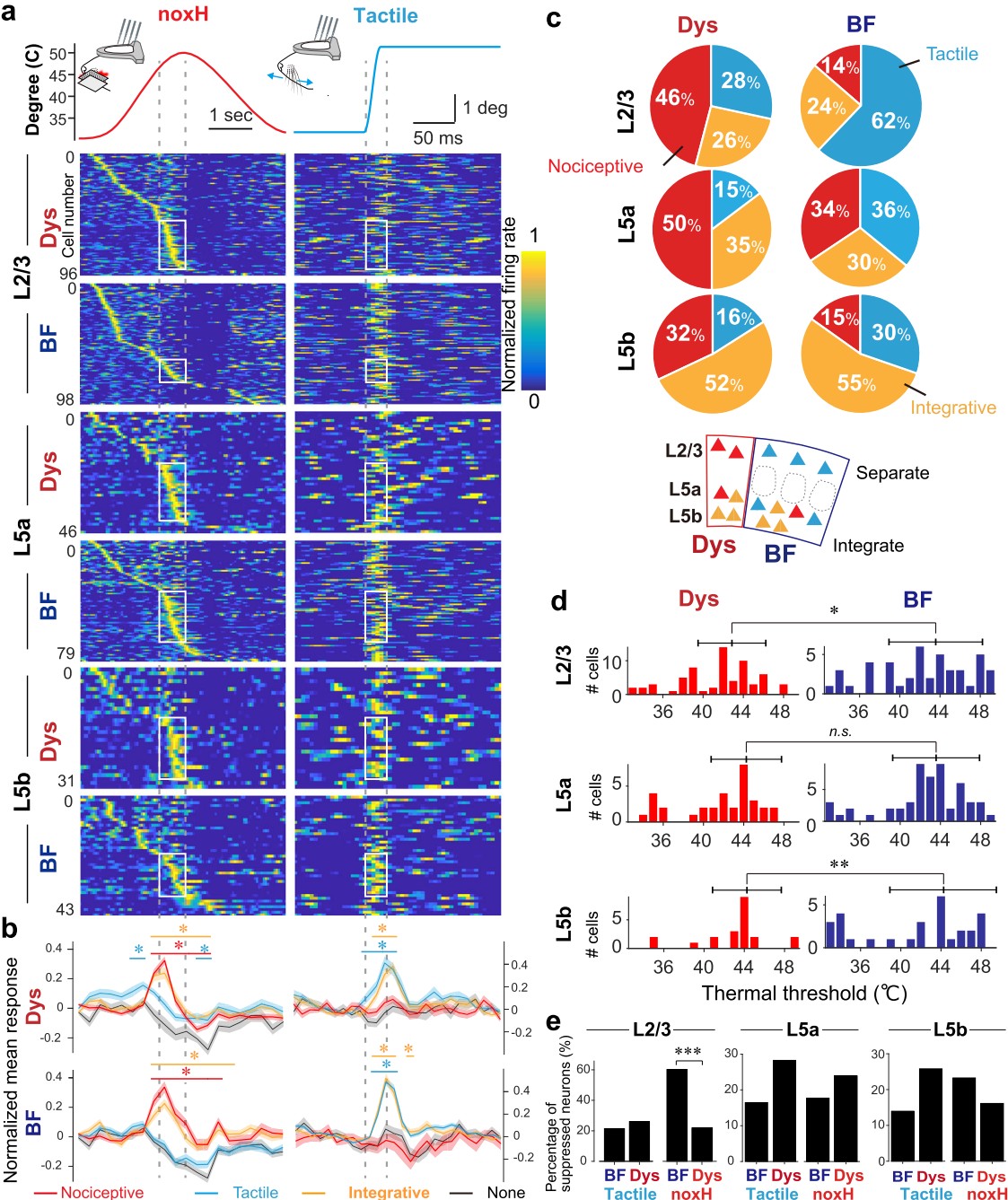

**Fig. 3 Separate processing and integration of nociception and tactile information in S1. a** PSTHs for the responses of the same neurons in L2/3, L5a, and L5b to noxious heat (noxH) (*left*) and tactile (*right*) stimuli. 50 ms/bin for the heat stimulus, and 5 ms/bin for the tactile stimulus. Each row was sorted by the peak response time for noxH, and the firing rate was normalized by the peak response (boxed areas) in each row. **b** Mean values from the histograms in panels a and b sorted according to the S/N to noxH and tactile stimuli: nociceptive, tactile, integrative (nociceptive and tactile), and non-reactive cells (none) (see Supplementary Fig. 3). *$P < 0.05$ versus non-reactive (two-sided Kruskal–Wallis test followed by Tukey's test). Shadings indicate SEMs. **c** Cell type distributions in L2/3, L5a, and L5b of each area and a summary diagram. **d** Distributions of the thermal thresholds determined from the temperature at which the neural responses reached 80% of the peak spike rate. In L2/3 and L5b, Dys neurons were tuned to noxious heat, whereas BF neurons responded to various temperatures (*$P = 0.013$, **$P = 0.005$, two-sample Ansari–Bradley test for dispersions). **e** Percentages of neurons for which responses were suppressed (S/N < 1). ***$P = 5.4 \times 10^{-8}$, 2 × 2 $\chi^2$ test between BF and Dys.

Therefore, we used the median S/N as the cutoff value for classification. According to this classification, the percentage of neurons of each type in each subregion was calculated (Fig. 3c). Consistent with the population PSTH (Fig. 3a), in the majority of neurons in L2/3, nociceptive information was processed separately in Dys while tactile information was processed in BF. Furthermore,

many tactile-type neurons in Dys L2/3 had relatively longer onset-latencies to tactile stimulation than those in BF L2/3 (Supplementary Fig. 4), suggesting that these neurons receive tactile information from BF. In contrast, the onset latency of integrative-type neurons in L2/3 of both regions was longer (Supplementary Fig. 4).

In the deeper layers, nociceptive and tactile information were integrated in the majority of neurons of both subregions (Fig. 3c, bottom).

Given that Dys and BF neurons in deeper layers responded mostly to noxious heat stimulation, we also examined neuronal responses to noxious mechanical stimuli[22]. We examined the evoked responses to a von Frey (10 g) noxious mechanical stimulus (Supplementary Fig. 5). The MUA of Dys and BF in L5b increased upon the application of a 10-g weighted von Frey filament (Supplementary Fig. 5a). The MUA of BF L5b and Dys L5b attained a maximum response when the von Frey filament bent (i.e., the whisker pad was stimulated at 10 g, Supplementary Fig. 5b). The steepness of the maximum responses to noxious stimulation increased in Dys L5b and BF L5b (Supplementary Fig. 5b), indicating that many neurons within the deeper layers also respond to noxious mechanical stimulation.

In the population PSTHs (Fig. 3a), the onset of the response to heat stimuli varied across neurons, especially in BF L2/3 and L5b, indicating that the neurons responded to various temperatures. Therefore, we examined the temperature sensitivity from the thermal threshold (80% maximum response) of neurons in response to the heat stimulus (Fig. 3d). L2/3 neurons in both regions began responding from innoxious temperatures (–44 °C). The distributions of the thermal threshold of Dys neurons, but not of BF neurons, were significantly accumulated around the noxious heat range in L2/3 ($P = 0.013$) and L5b ($P = 0.0051$) (Fig. 3d). These data suggest that Dys neurons are well-tuned to noxious heat temperature, whereas neurons in BF L2/3 and L5b encode cutaneous temperatures[28,29].

The average PSTHs (Fig. 3b) show that the neuronal activities of tactile and non-reactive neurons were suppressed by noxH. Because the S/N of these neurons was <1, we estimated the percentage of neurons suppressed by noxH in each region. In BF L2/3, 60% of neurons were suppressed by noxH (Fig. 3e). This percentage is significantly larger than that in Dys (22%, $P = 5.4 \times 10^{-8}$). These data may explain the neural mechanism underlying deficits in tactile perceptual discriminative capacity under the condition of pain[30,31].

In summary, the data show that nociceptive information is processed separately from tactile information in the superficial layer of Dys and BF and tends to be integrated in deeper layers. The difference in thermal thresholds indicates that Dys processes noxious heat input, and BF is responsible for temperature coding.

**Dys is involved in neuropathic pain.** We next examined whether Dys activity involves pain processing under pathophysiological conditions. It has been proposed that S1 is activated under chronic pain[12,13,32,33] and that peripheral nerve fiber injury causes somatotopic reorganization and mechanical hypersensitivity[34–39]. Therefore, we applied a trigeminal neuralgia model by ligation of the infraorbital nerve (ION) and investigated how spatial cortical activity changes in S1 during the development of mechanical allodynia. Toward this aim, we used an absorbable surgical thread (see Methods), which enabled us to observe the mechanical allodynia state and recovery state in the same animal.

We first examined the time course of the development of mechanical allodynia by the ION ligation, which was evaluated by the escape threshold in the von Frey filament test. The mice showed allodynia when the tensile strength of the surgical thread was ~50% and recovered when the tensile strength was reduced to 0% of the maximum strength (Supplementary Fig. 6).

Next, we recorded intrinsic signals induced by whisker stimulation continuously from the same animal before and during the mechanical allodynia state (POD7) and recovery state

(POD21, Fig. 4a, b). Evident intrinsic signals in BF induced by whisker stimulation were observed before ligation (Fig. 4b, left). At the allodynia state (POD7), the signal in BF disappeared but was enhanced in the adjacent region (Fig. 4b, middle), although the signal contrast was relatively low (Fig. 4b). At the recovery state (POD21), the signal in BF reappeared (Fig. 4b, right). After a recording series, we confirmed that the adjacent region of BF, where the signal was detected at POD7, was Dys in a subsequent histological analysis (Fig. 4c and Supplementary Fig. 7).

To confirm Dys neuronal activity during the ligation and recovery, we recorded the stimulus-evoked MUA in Dys and BF at POD7 and POD21 in individual animals (Supplementary Fig. 8a, b). The balance of activity between BF and Dys was altered (Supplementary Fig. 8a, b). Dys activity was higher than in BF in some animals during the ligation at POD7 (Supplementary Fig. 8a); moreover, BF activity recovered at POD21 (Supplementary Fig. 8b). Intrinsic signal imaging results from the spatial summation of the neuronal activity and electrophysiological recordings of localized neuronal activity may therefore underestimate changes in Dys activity. Accordingly, we also counted c-Fos positive neurons in mice after the mice had explored an enriched environment[40,41] (See Methods, Supplementary Fig. 9a). Consistent with the intrinsic signal imaging results, the number of c-Fos-positive neurons was significantly increased in Dys but decreased in BF at POD7. However, at POD21, the number of c-Fos-positive neurons in BF was recovered (Supplementary Fig. 9b, c). Thus, the activated region changed dynamically according to the extent of the peripheral nerve injury, and the neuronal activity in Dys was increased in the pain-like condition.

**Dys is involved in generating pain-like behavior.** Finally, we examined whether Dys is involved in pain-like behavior, such as escape from harmful stimuli. For this, we monitored the behaviors of head-restrained animals freely moving on a spherical treadmill[42] in response to an innoxious or noxious infrared (IR) laser applied to the left whisker pad (Fig. 5a). Application of the IR laser for 500 and 1500 ms increases the skin temperature to 39 °C and 52 °C, respectively[43], which we thus refer to as innoxious heat (innH) and noxH, respectively. In response to noxH, the traveling speed increased until 5 s after the onset of the IR laser stimulus and the running direction changed toward the opposite side to the noxious input as the mice attempted to escape from the noxious input (Fig. 5b–d and Supplementary Fig. 10a, Direction). The animals also exhibited eye blinking and tightening (Supplementary Fig. 10a), which are considered expressions of pain[44–46]. Thus, this system was suitable for quantifying pain-like behaviors in response to noxH.

Using this system, we examined the effect of modulating Dys activity on pain-like behavior. We monitored pain-like behaviors induced by noxH during the optogenetic suppression of various cortical areas, including Dys (Fig. 5e). For this experiment, we did not assess eye blinking or tightening, which involve reflex actions via the brainstem and cerebellum[47]. We used a transgenic line that expresses channelrhodopsin-2 (ChR2)-EYFP in parvalbumin (PV)-expressing interneurons (PV-Cre × Ai32) and photoactivated the PV interneurons to inhibit cortical pyramidal neurons locally[48–50] (Fig. 5e). Since Dys is a narrow region (approximately 0.4 mm) that is adjacent to BF (tangential section, Fig.4c), stimulating Dys while excluding BF or the paw region is challenging. We therefore changed the laser stimulation positions at six points and then compared the behaviors[48,50]. Photoinhibition at position 3 (P3) and P4, which mainly covered Dys and partially the paw region or BF, significantly decreased the escape speed in response to noxH (1.5 to 2 s at P3 and P4, $P < 0.05$; and 2 to 2.5 s at P4, $n = 6$, Fig. 5f). By contrast, the photoinhibition of

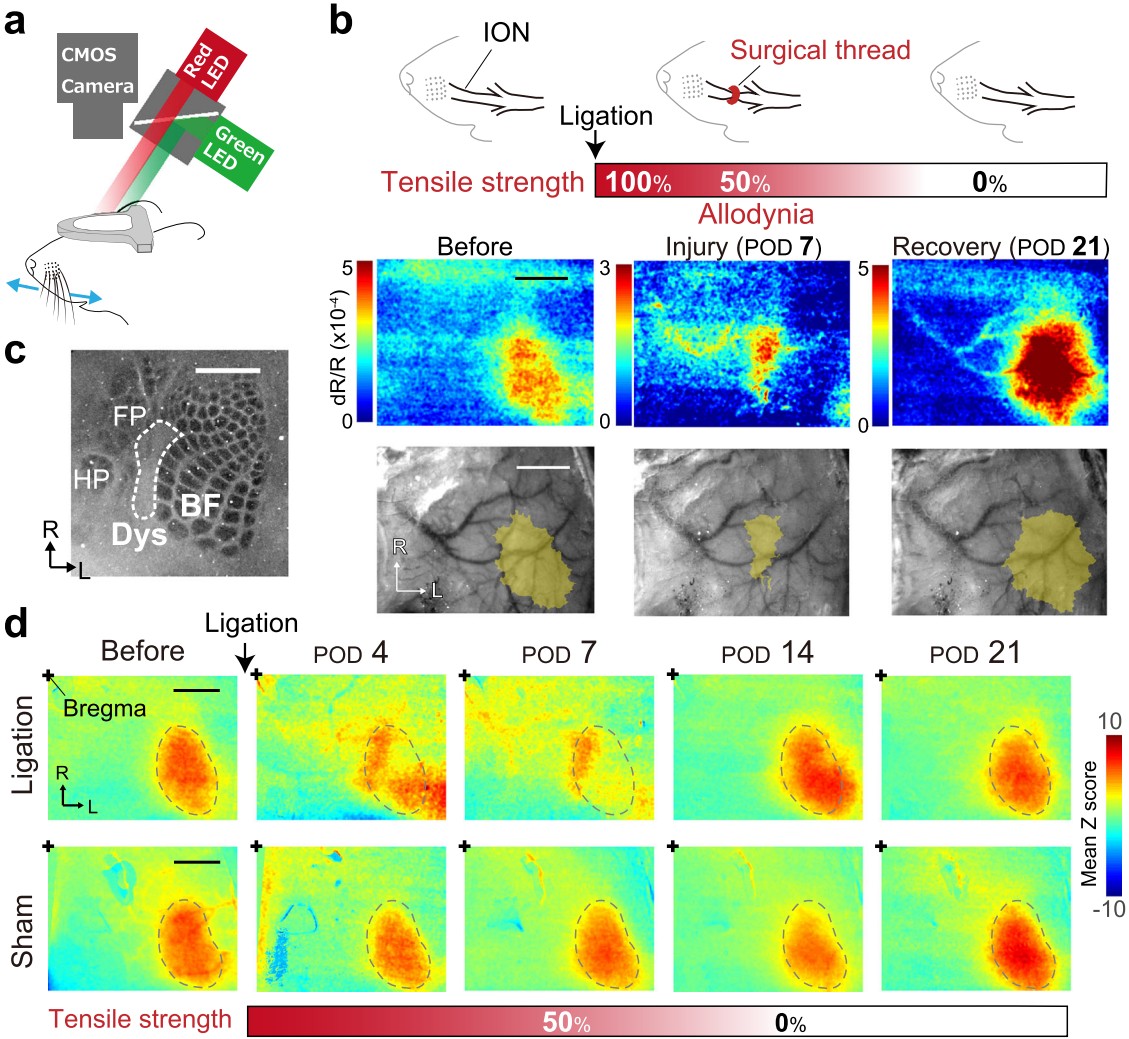

**Fig. 4 Tactile activated region is shifted from the barrel field to the dysgranular region during peripheral nerve injury. a** Schematic showing intrinsic signal imaging during whisker stimulation by the piezo device. A red LED was used for intrinsic signal imaging, and a green LED was used to obtain the vessel pattern of the brain surface. **b** Top, Schematic of the time course for infraorbital nerve (ION) ligation by an absorbable surgical thread. Middle, Typical examples of the intrinsic signal images of an animal. *Bottom*, Overlaid images of the signal region (yellow) and vessel pattern used for the alignment. **c** Cytochrome c oxidase staining of a tangential section of S1 L4. FP, forepaw; HP, hindpaw. **d** Averaged z-scored images (ligation group, $n = 5$; sham group, $n = 3$). The dotted areas indicate BF activated by the whisker stimulation before ligation. R, rostral; L, lateral; POD, postoperative day. Scale bars, 1 mm.

other S1 regions, such as BF or paw regions, had no effect ($P > 0.05$, Supplementary Fig. 11). Similar trends were observed for the maximal speed ($P < 0.01$ at P3 and P4, not significant at other positions, Fig. 5g), and for changes in the escape direction and distance (Supplementary Fig. 10b). In summary, we confirmed that photoinhibition at P3 and P4 suppressed the escape behavior of mice running in the opposite direction of the noxious input. In control mice that did not express ChR2, blue laser stimulation did not affect the escape speed (Supplementary Fig. 12). These results indicate that the optogenetic local suppression of Dys neurons reduced noxH evoked pain-like behaviors.

To support this observation, we photoactivated the thalamocortical fiber to activate Dys. Because medial posterior nucleus (POm) neurons are thought to bring nociceptive information from the thalamus to S1[11], we confirmed the connection from POm neurons to Dys by retro- and anterograde tracers (Supplementary Fig. 13)[16] and utilized mice with the virus-induced expression of ChR2 (AAV9-hSyn-ChR2(H134R)-EYFP) in POm neurons (Supplementary Fig. 14a)[16]. Using these mice,

we observed that the photoactivation of Dys in response to innH resulted in treadmill speed profiles resembling those in response to noxH (Supplementary Fig. 14b). Similarly, the maximal speed with innH paired with Dys photoactivation was almost the same as that in response to noxH ($P = 0.99$, Supplementary Fig. 14c). Similar trends were observed for the escape direction and distance (Supplementary Fig. 14d, e). These results show that ChR2-mediated photoactivation of Dys enhanced pain-like behaviors. In control mice lacking ChR2 in POm, blue laser stimulation had no effect (Supplementary Fig. 14f–j). In summary, the optogenetic studies demonstrate that Dys inhibition reduces pain-like behaviors, while Dys activation induces them in response to innH, revealing the role of Dys in generating pain-like behavior.

## Discussion
This study reveals that Dys shows high sensitivity to noxious inputs, whereas BF prefers tactile inputs. In particular, nociceptive information is processed separately from tactile information in Dys L2/3. Dys is also responsive to neuropathic pain. Reflecting the spatially distinct representation of nociception, the

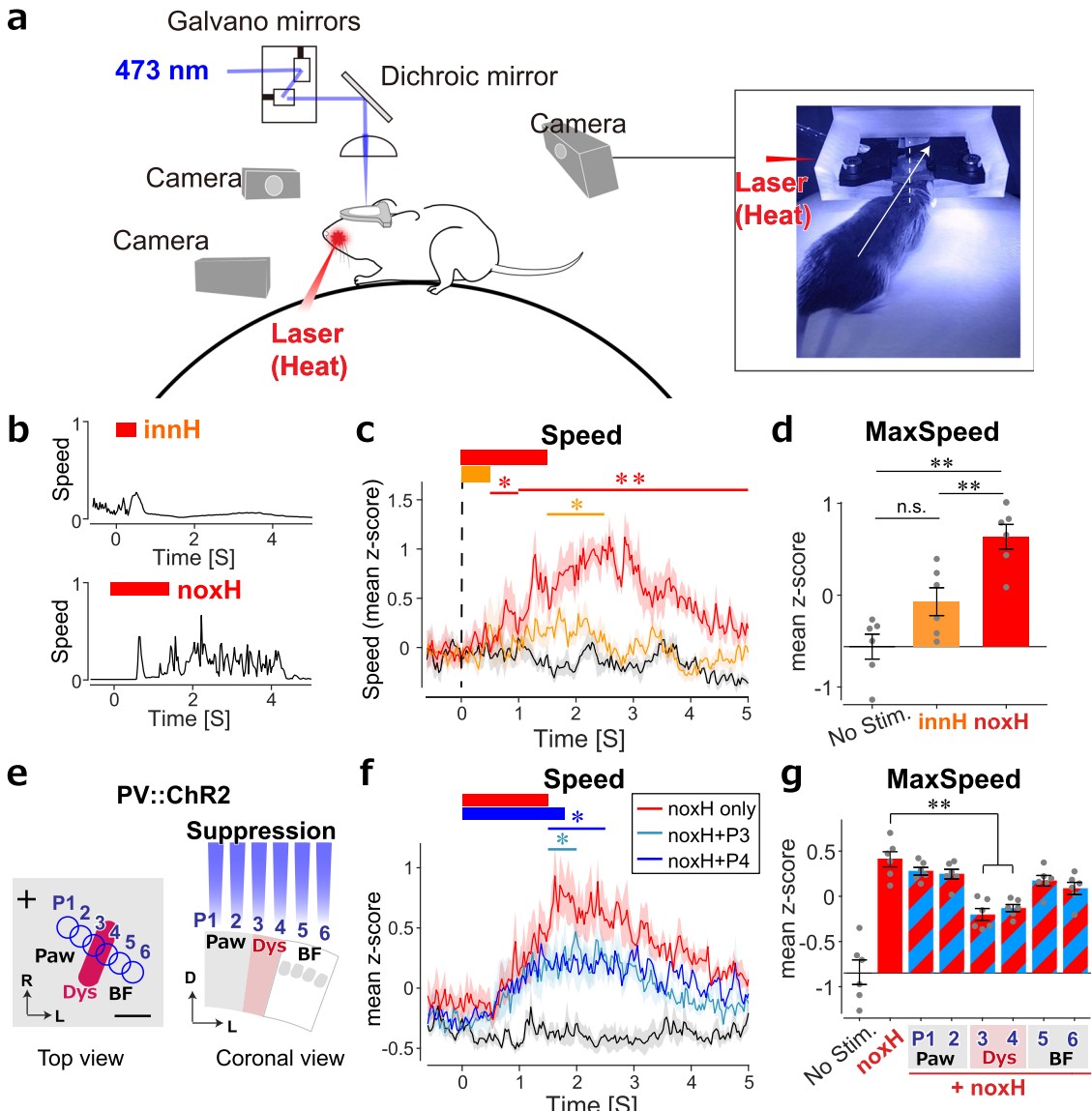

**Fig. 5 Dysgranular region activity induces nocifensive escape behavior. a** Schematic for monitoring escape behaviour induced by applying an 808-nm infrared (IR) laser to the left whisker pad of a head-restrained freely moving animal on a spherical treadmill. Motion direction and speed were monitored by a camera at the back, and facial expression and forelimb movement were monitored by side cameras. **b** Examples of escape speed in response to 500 and 1500 ms IR stimulations, corresponding to 0.09 (innoxious heat, innH) and 0.27 J/mm$^2$ (noxH), respectively. **c** Average speed profiles with (innH, orange; noxH, red) and without (black) IR stimulation ($n = 6$ mice). *$P < 0.05$, **$P < 0.01$. **d** Maximum speeds induced by IR stimulation ($n = 6$ mice, n.s., not significant; **$P = 3.1 \times 10^{-6}$ (No Stim. v.s. noxH), 0.0055 (innH v.s. nox H); one-way ANOVA followed by the Tukey–Kramer test.). **e** Scheme for silencing six S1 positions (430 μm apart) via 473-nm photoactivation of channelrhodopsin-2 (ChR2) during noxH stimulation. Scale bar, 1 mm. +, bregma; R, rostral; L, lateral; D, dorsal. **f** Average speed profiles show that optogenetic suppression at P3 (blue) or P4 (cyan) significantly reduced the escape speed to noxH ($n = 6$ mice). *$P < 0.05$. **g** Mean $z$-scores of the maximum speed ($n = 6$ mice). **$P = 2.3 \times 10^{-5}$ (noxH v.s. P3), 2.0 × $10^{-4}$ (noxH v.s. P4); one-way ANOVA followed by the Tukey–Kramer test. Shading and error bars indicate SEMs for mice.

optogenetic suppression of Dys activity reduced noxious heat-evoked pain-like behaviors, whereas the same manipulation in BF showed no behavioral effects. These findings indicate that Dys is involved in nociception and in the generation of pain-like behaviors.

**Separation and interaction of nociceptive and tactile processing.** Previous studies have reported that nociresponsive cortical neurons are located in the deeper layers of BF[11,22,23] and Dys[24], but the functional differences between the two subregions remain unknown. Here, we demonstrate that Dys and BF have segregated roles in nociceptive and tactile processing, especially in L2/3. Our

neurophysiological data further show that noxH input suppresses neural activity in BF L2/3. This suppression may be the neural basis for disrupting the acuity of tactile sensation during pain[30,31].

In contrast to L2/3, the segregation of nociceptive and tactile information processing was less clear in L5b, with larger proportions of integrative neurons in both regions. However, the preference for tactile stimuli in L5 BF neurons was maintained; BF neurons showed a higher preference to tactile input than Dys neurons (Fig. 2c). Because L5 neurons are suggested to modulate nociceptive inputs through the corticofugal pathway[14,15], BF L5 neurons might relate to touch-induced pain relief under normal conditions[51], while Dys L5 neurons might contribute to mechanical allodynia (Fig. 4).

**Selective intervention of S1 activity**. The notable finding in the present study is that Dys, but not BF, is involved in generating escape behavior (Fig. 5). For appropriate escape behavior, the animal needs to run in a direction away from the noxious input (Fig. 5d and Supplementary Fig. 10a). Dys inactivation by the optogenetic activation of local PV interneurons reduced the escape speed and disrupted the directional selectivity to escape from noxious inputs (Fig. 5g and Supplementary Fig. 10b). Several lines of anatomical evidence suggest that Dys is intimately related to motor function and pain processing. Dys, for example, receives proprioceptive inputs via POm[16,20,21] and projects to the primary motor cortex[27]. Dys also projects to the anterior cingulate cortex[27], which integrates sensory and affective pain and modulates pain-like behavior[12,13]. Moreover, Dys has a neuronal connection with the secondary somatosensory cortex[27], which is involved in the processing of nociceptive information[52,53]. It is therefore likely that Dys-mediated neuronal networks promote appropriate escape behavior by integrating proprioception and nociception processing.

The photoactivation of thalamocortical fibers projecting from POm to Dys induced escape behavior (Supplementary Fig. 14), confirming the findings from the optogenetic inhibition of Dys (Fig. 5). Since neurons in the POm project to Dys L4 and BF L5[16,54,55] (Supplementary Fig. 13b), it may not be possible to completely exclude the contribution of BF. BF activation reduces pain-like behavior via the feedforward inhibition of corticotrigeminal axons from L5[14]. Moreover, the descending projections from Dys terminate in areas distinct from those from S1 cutaneous regions, such as BF[56]. Considering these reports, in the current study, the photoactivation of POm-S1 fibers that induce pain behavior are a consequence of POm-Dys fiber activation. These findings suggest that Dys and BF produce different effects on escape and pain-like behaviors.

**Possibility of S1 intervention for pain relief based on S1 functional structure**. It has been suggested that controlling S1 activity modulates pain thresholds[14,33]. Given that the manipulation of Dys activity controls pain-like behavior (Fig. 5), selective inhibition of the nociresponsive region in S1 may be a therapeutic target in clinical pain management without affecting other functions of S1. Cerebral cortical lesions in S1 cause permanent pain relief (reviewed in Whitsel et al[57].). Neurons in non-human primate area 3a, a dysgranular region based on its cytoarchitecture[57], respond to noxious[8,57,58] and proprioceptive inputs[59]. These observations and phylogenetic relationships[60] suggest that primate area 3a is an evolutional homolog to rodent Dys. Moreover, human area 3a may play a central role in pain perception[61]. Thus, human area 3a may be one of the ideal targets of intervention for pain relief. However, area 3a is buried in the fundus of the central sulcus in humans[62]. Therefore, the selective intervention of area 3a is challenging and needs to be resolved for the effective treatment of chronic pain in the future[57,58]. Nevertheless, our results strongly suggest that the distinctive nociceptive region in S1 is a potential therapeutic target for pain relief.

## Methods

**Animals and surgery**. All surgical procedures and postoperative care were performed according to the guidelines of the Animal Care and Use Committee of Tokyo Women's Medical University. All animal experiments were approved under number AE19-109. C57BL/6 N (Sankyo Lab. Service Corp., Tokyo, Japan), PV-Cre (JAX stock #008069), and Ai32 (Rosa-CAG-LSL-ChR2[H134R]-EYFP-WPRE; JAX stock #012569) mouse lines were used in this study. PV-Cre mice were crossed with Ai32 mice, and the resulting mouse line was designated PV-ChR2. Male mice of 8 weeks or older in age were used. The animals were group-housed in a cage maintained at 23 °C ± 1 °C, 50 ± 15% humidity with a 12 h light/dark cycle, and all behavioral tests were performed during the dark period. Every effort was made to minimize the number of mice used and their suffering in this study. For surgical

procedures, each animal was anesthetized with an intraperitoneal injection of ketamine (100 mg/kg body weight) and xylazine (16 mg/kg body weight), and isoflurane was supplemented to maintain the anesthesia. Lidocaine was applied subcutaneously at the incision site and to the wound margins for topical anesthesia. For intrinsic signal optical imaging, electrophysiological recordings, and behavioral testing on a spherical treadmill, a custom-built headplate was attached to the skull with dental acrylic clear resin (Super-Bond; Sun Medical, Shiga, Japan). The head plate-implanted animals were returned to the home cage and allowed to recover from the surgery for at least 4 days.

**Electrophysiological recording**. For electrophysiological recordings from Dys and BF, each mouse was anesthetized with isoflurane (0.4–0.8%) supplemented with an intraperitoneal injection of chlorprothixene hydrochloride (2 mg/kg body weight)[63]. Because the nociceptive response depends on the level of anaesthesia[64], the respiration rate (70–120 cycles/s) was monitored to control the level of anesthesia.

We first identified the border between Dys and BF according to intrinsic signal imaging by whisker stimulation (See "Intrinsic signal optical imaging"). Then, 32-channel four-shank electrodes (A4x8 or Buzsaki32; NeuroNexus, Ann Arbor, MI, USA) stained with DiI (V22885, Thermo Fisher Scientific, Waltham, MA, USA) were inserted into both Dys and BF. We simultaneously recorded neural activity from each region with the receptive fields of the E-row whiskers. To record neurons with receptive fields covering the E-row whiskers and the whisker pad around the E-row[65], at least two shanks were positioned in the E-row barrel and Dys adjacent to the E-row barrel (Fig. 1a).

Raw electrical signals were amplified and digitized at 40 kHz (Plexon, Dallas, TX, USA) and then processed for spike sorting. The spike sorting comprised automated spike detection and clustering using Klusta (ver. 3.0.16) followed by manual sorting using Kwik GUI (v1.0.9)[66]. First, noise artifacts determined from the waveform were extracted. Second, multiunit activity (MUA) was determined from the waveform with low amplitude and without a refractory period (>2 ms) in the auto-correlograms. Third, after merging and/or splitting the clusters using auto- and cross-correlograms and principal component features, single-unit activity (SUA), which has a clear refractory period (>2 ms) in the auto-correlograms, or MUA was determined. To estimate the depth of the recorded neurons, the maximum amplitudes of the waveforms from each probe were compared and determined for the nearest probe for each SUA/MUA. For the MUA analysis, SUA was included (n = 8 animals, 128 probe sites simultaneously recorded for the MUA analysis).

For the electrophysiological data analysis of brushing on the left whisker pad (Supplementary Fig. 2), von Frey stimulation (Supplementary Fig. 5), and stimulation during ION ligation (Supplementary Fig. 8), raw electrical signals were amplified and digitized at 30 kHz (Open Ephys[67], Open Ephys GUI, v0.5.0) and then processed for spike sorting. Spike sorting comprised automated spike detection and clustering using Kilosort (v2.0, https://github.com/cortex-lab/KiloSort) followed by manual sorting using Phy (v2.0b1, https://phy.readthedocs.io/en/latest/). For the MUA analysis at POD7 and POD21 (Supplementary Fig. 8), we used the same 32-channel, 4-shank electrode (Buzsaki32) across each recording. We then selected the neighboring two shanks in Dys and BF and compared the total number of spikes recorded from each shank.

After the recordings, the mice were deeply anesthetized with sodium pentobarbital (60 mg/kg body weight, intraperitoneally) and transcardially perfused with a fixative solution (4% paraformaldehyde and 0.2% picric acid in 0.1 M phosphate buffer). The brains were removed and cut coronally into 50 μm sections. The sections were incubated overnight with a guinea pig polyclonal antibody against VGluT2 (guinea pig; 1:500, MSFR106290, Nittobo Medical Co., Ltd., Tokyo, Japan) followed by NeuroTrace 435/455 (1:100; Thermo Fisher Scientific, Waltham, MA, USA) to identify the recording sites. Images were captured with an upright microscope (AxioScope.A1, Zeiss, Germany) with a cooled-CCD camera (RS 6.1 s, QSI, Cambridge, England) with μManager (ver. 1.4, https://micro-manager.org/).

**Calculation of S/N for noxious and tactile stimuli**. We calculated the S/N to identify the area or cell properties and assessed selectivity to noxious heat and tactile stimuli. A Peltier device (4.2 mm × 4.0 mm) was placed on the left whisker pad to apply the heat stimulus. To ensure heat transfer to the skin, tiny hairs between whiskers were removed by hair removal foam, leaving the whiskers intact. For the S/N of the noxious heat response, the response for 500 ms at 45–50 °C for the Peltier device (beige shading, Fig. 1e) was counted as the S response. The innocuous heat response (for 1000 ms until 45 °C, corresponding to 33–45 °C, grey shading, Fig. 1e) was counted as the N response. This definition helped to select noxious heat-selective neurons by excluding temperature-coding neurons.

For tactile stimuli, the response for 30 ms after the onset of the whisker deflection (blue arrows, Fig. 2a, right) was counted as the S response. The response for 60 ms until the onset of the whisker deflection was counted as the N response (Fig. 2a). All deflections were used for this calculation (Fig. 2b). This definition helped to detect how the neuron responds to sequential whisker deflection[68].

**Onset detection of von Frey filament stimulation**. A von Frey filament (10 g) was applied to the left whisker pad as a stimulus for mechanical pain. All whiskers and tiny hairs between whiskers on the left whisker pad were removed to detect application of the 10-g von Frey filament to the skin. All stimulation trials were monitored by a high-speed camera recording at 200 Hz (XiQ, Ximea GmbH, Münster, Germany). The region of interest was placed on the whisker pad, and the bend of the skin was calculated (color plots in Supplementary Fig. 5a). A saturation point of bending of the skin identified the bending onset of the von Frey filament.

**Capsaicin injection and c-Fos immunofluorescence**. Capsaicin (Wako Pure Chemical Industries, Ltd., Osaka, Japan) was dissolved in 100% ethanol and 7% Tween 80 in saline (10 mM). A solution containing 100% ethanol, 7% Tween 80, and saline was used as the vehicle[69]. The animals were transferred in their home cages from the animal facility and anesthetized with 2% isoflurane. After the injection of capsaicin (50 μL) or vehicle (50 μL) into the left whisker pad, the mice were placed back into their home cages and returned to the animal facility.

Three hours after the injection, mice were deeply anesthetized with isoflurane (2–3%, 5 min) and perfused transcardially with a prefixative solution (250 mM sucrose and 5 mM MgCl$_2$ in 0.02 M phosphate-buffered saline, pH 7.4) containing heparin sodium salt (30–40 units/mL) followed by a fixative solution (4% paraformaldehyde and 0.2% picric acid in 0.1 M phosphate-buffered solution). Brains were postfixed at 4 °C for 12–16 h in fresh fixative and coronally cut into 40-μm-thick sections with a vibratome (VT1000S, Leica Microsystems Inc, Wetzlar, Germany). Sections were incubated at 20–25 °C for 12–16 h with a mixture of rabbit polyclonal antibody against c-Fos (1:2000; 226 003, Synaptic Systems GmbH, Göttingen, Germany), a guinea pig polyclonal antibody against VGluT2 (1:500; MSFR106290, Nittobo Medical Co., Ltd., Tokyo, Japan), a goat polyclonal antibody against calbindin D-28K (1:500; MSFR100410, Nittobo Medical Co., Ltd.), and a mouse monoclonal antibody against NeuN (1:500; MAB377, Merck, Darmstadt, Germany) in 0.05 M phosphate-buffered saline (PBS) containing 10% normal donkey serum and 0.3% Triton X-100. After washing 2–3 times with PBS, sections were further incubated at 20–25 °C for 2–3 h with secondary antibodies conjugated to Alexa Fluor Plus 555 (for c-Fos, 1:500; A32794, Thermo Fisher Scientific, Waltham, MA, USA), Alexa Fluor 647 (for VGluT2, 1:500; 706-605-148, Jackson ImmunoResearch, West Grove, PA, USA), Alexa Fluor 488 (for calbindin, 1:500; A11055, Thermo Fisher Scientific), and Alexa Fluor Plus 405 (for NeuN, 1:500; A48257, Thermo Fisher Scientific) in 0.05 M PBS containing 10% normal donkey serum and 0.3% Triton X-100. After washing 2–3 times with 0.1 M phosphate buffer, sections were mounted on glass slides, sealed with SlowFade Diamond antifade mountant (Thermo Fisher Scientific), and coverslipped. Images were acquired using a fluorescence microscope (BZ-X 810, Keyence, Tokyo, Japan). The number of c-Fos-expressing cells in each column of BF and Dys was counted manually. The number of NeuN stained neurons in the same regions of interest was also counted. Manual counting was performed by a blinded independent observer.

**c-Fos immunohistochemistry using DAB**. After the brains were postfixed, 50-μm-thick slices were made, and alternate slices were reacted with cytochrome c oxidase to identify BF, and with c-Fos immunohistochemistry. For c-Fos immunohistochemistry, the slices were processed with 1% H$_2$O$_2$ in phosphate buffer to deactivate the intrinsic peroxidase and then incubated with anti-c-Fos antibody (1:10000, rabbit; Merck KGaA, Darmstadt, Germany) in 10% normal goat serum in phosphate-buffered saline with 0.3% Triton X-100 at 4 °C overnight. The slices were then incubated with biotinylated goat anti-rabbit IgG antibody (1:200; Vector Laboratories, Burlingame, CA, USA) and reacted with avidin-biotin-peroxidase complex (ABC kit; Vector Laboratories). The slices were incubated in DAB solution (0.02% DAB, 0.3% nickel ammonium sulfate in Tris-buffered saline) and 1% H$_2$O$_2$ for visualization. Images were acquired with an upright microscope with a CCD camera (DP70; Olympus, Tokyo, Japan) using Olympus DP Manager (ver. 3.1.1.208, Olympus). Neurons expressing c-Fos were counted using a custom-written MATLAB (MathWorks, Natick, MA, USA) program (Supplementary Software 1) with edge detection by a Sobel filter followed by binarization. The number of c-Fos-expressing cells was manually counted in three slices by a blinded independent observer; this individual was not involved in the automatic counting and confirmed that the tendency of c-Fos expression in each layer did not differ across regions.

**Anterograde/retrograde labelling**. For retrograde labeling, cholera toxin subunit B conjugated with Alexa Fluor 555 (0.2%) or Alexa Fluor 488 (1%) was applied after intrinsic signal imaging to identify Dys and BF, respectively. Three days later, the mice were deeply anesthetized with sodium pentobarbital (60 mg/kg body weight, intraperitoneally) and transcardially perfused with a fixative solution (4% paraformaldehyde and 0.2% picric acid in 0.1 M phosphate buffer). The brains were removed and cut coronally into 50 μm sections. Sections were incubated overnight with a guinea pig polyclonal antibody against VGluT2 (1:500; MSFR106290, Nittobo Medical Co., Ltd., Tokyo, Japan) followed by NeuroTrace 435/455 (1:100; Thermo Fisher Scientific, Waltham, MA, USA).

For anterograde labeling, biotinylated dextran amine (molecular weight: 10,000, 10% in saline; Thermo Fisher Scientific) was injected into the POm (1.7 mm posterior to bregma and 1.3 mm lateral to the midline). Seven days later, the mice were deeply anesthetized, and the brain sections were cut as described above.

Sections were incubated overnight with VGluT2 antibody (guinea pig; 1:500, MSFR106290, Nittobo Medical Co., Ltd., Tokyo, Japan) followed by Alexa Fluor 594-conjugated anti-guinea pig antibody (1:500; 706-585-148, Jackson ImmunoResearch), Alexa Fluor 488-conjugated streptavidin (1:500; S11223, Thermo Fisher Scientific), and subsequently with NeuroTrace 435/455 (1:100; S21479, Thermo Fisher Scientific).

**Intrinsic signal optical imaging**. At least 4 days after the head plate implantation, intrinsic signal imaging was performed to measure the responses from BF. The mouse was anesthetized as described above in "Electrophysiological recording." The respiration rate and heart rate were monitored via a video-based respiration monitor with a 30 Hz web camera (C920; Logitech, Lausanne, Switzerland) and a custom-made acceleration monitor on the back of the animal. Intrinsic signal images were obtained using microDisplay software (ver. 5.2.3.1, Silicon Software GmbH, Germany) by a CMOS camera (MV1-D1024E-160-CL; Photonfocus, Lachen, Switzerland) with the tandem lens of an achromatic doublet (Thorlabs, Newton, NJ, USA) and long-pass and short-pass filters (BLP01-488R-25 and FF01-650/SP-25; Semrock, Rochester, NY, USA). Frames were acquired at a rate of 20 Hz, and the frame size was 600 × 500 pixels and represented 5.5 × 4.5 mm of the cortical area. The brain surface was illuminated by a green LED (M530L3; Thorlabs) to obtain the vessel pattern or a red LED (M617L3; Thorlabs) to obtain the intrinsic signal. The red LED luminance was constant between each imaging. The respiratory rate (70–120 cycles/s) was monitored to track the depth of anesthesia. Images were recorded through the skull covered with dental acrylic clear resin. The dental acrylic resin was covered with a nail topcoat and silicone immersion oil (Olympus) to reduce glare.

Whisker stimulations as tactile stimuli were generated using a piezoelectric device as described previously[68]. To visualize the cortical response to tactile stimuli, we calculated the reflectance ratio in each frame (dR/R, where dR is the difference of reflectance R from the base image that is the average from 20 frames taken before the stimulus onset). To map the change in cortical activity, images taken on different experimental days were aligned according to vessel patterns using a custom-written MATLAB program. For population analysis, we calculated the z-scored image from dR/R using the following equation:

$$z-score(pixel) = \frac{\frac{dR}{R}(pixel) - mean\left(\frac{dR}{R}(all\ pixels)\right)}{SD(all\ pixels)} \tag{1}$$

where SD is the standard deviation. The positions of the bregma were used for the alignment between animals.

**ION ligation and behavioral assay using von Frey filaments**. To produce a neuropathic pain model, the ION was tightly ligated by a surgical thread (Vicryl Rapide; Ethicon, Bridgewater, NJ, USA). This thread enables us to observe changes in BF activity during both nerve injury and recovery, as the tensile strength of the thread gradually reduces inside the body (within 2–3 weeks); the tensile strength is reduced to almost 50% at POD7 and 0% at POD21, corresponding to the phases of nerve injury and recovery, respectively. For the behavioral assay of neuropathic pain, the animals were trained to enter a 50-mL tube with a custom-made tube holder. Behavioral training began after the mice had restricted access to water (1 mL/day) at least 7 days before the left ION ligation or sham operation. The water was restricted to 1.5 mL for 1 day during the behavioral experiment. The animals were trained to enter the tube and keep their snouts protruding through a hole to drink the water. While the animals were drinking, the left whisker pad was stimulated by the von Frey filaments (1.4, 2, 4, 6, 8, and 10 g; Ugo Basile, Varese, Italy) to measure the escape threshold[70]. During the stimulation, visual information was blocked by a black cover (Supplementary Fig. 6).

**Environmental enrichment**. At 7 or 21 days after the ION ligation and 7 days after the sham operation, the mice were placed into an enriched environment for 1 h to increase whisking while they explored several objects (Supplementary Fig. 9)[40,41]. Then, the mice were perfused with 4% paraformaldehyde with picric acid followed by c-Fos immunohistochemistry using the DAB protocol described above.

**Measurement and analysis for pain behavior on a spherical treadmill**. Mice were first trained to enter a tube to obtain a water reward for 3–4 days. Mice were then head-fixed and free to run on a spherical treadmill (Ø 30 cm) under white noise[71] for 3–4 days. After this, the behaviors of the mice towards the IR laser stimulus on the left whisker pad were monitored. For the noxious heat stimulus, an IR diode laser (Ø 1 mm on the whisker pad, wavelength of 808 nm, SSL-808-1000-10TM-D-LED; Shanghai Sanctity Laser Technology Co., Ltd., Shanghai, China) was used. The stimulus duration was set to 500 or 1,500 ms, corresponding to 0.09 and 0.27 J/mm$^2$, respectively, to increase the skin temperature to 39 °C or 52 °C[43]. At the start and end of each session, the animals obtained a water reward on the treadmill but not during the stimulus sessions. Each session was composed of various stimulus conditions, and each condition was randomly chosen and presented to the animal five times in one session. The animals were imaged with three cameras at 30 Hz (the IR filter on the CMOS sensor was removed; C922, Logitech, Lausanne, Switzerland) set behind and to each side of the animal to record behaviors such as escape direction, speed, moving distance, eyeblink, and left forelimb movement. These parameters were

analyzed by calculating the difference of the region of interest of each parameter frame by frame. These parameters were calculated into z-scores for comparisons among the animals by a custom-written MATLAB code (Supplementary Software 2). If the animal ran continuously on the treadmill prior to the onset of the stimulus, the effect of the stimulus was masked. These animals tended to run continuously regardless of the stimuli. Therefore, no difference in the maximum speed between trials could be observed, and these sessions were excluded from the analysis. At least two sessions were used to calculate z-scores. A notch filter (808 nm OD4 notch filter, 86-702; Edmund Optics) was placed in front of the left side camera to prevent sensor white-out and to identify the precise position and size of the IR laser stimulation. Custom-made LED illuminators (940 nm) were placed in front of each camera to illuminate the animals.

**Optogenetic inactivation and activation of Dys.** To activate the thalamocortical fibers from POm into Dys, a virus (AAV9-hSyn-hChR2(H134R)-EYFP) was injected into the right POm (1.7–1.9 mm caudal and 1.2 mm lateral to bregma; depth, 2800 and 3000 μm from the brain surface; 50 nL at each depth) (QSI; Stoelting, Wood Dale, IL, USA). A blue laser (473 nm; Lasos, Germany) was coupled to an optic fiber cable (Ø 200 μm; Thorlabs). The output of the optic fiber and the surface of the cortex were placed on conjugate planes using a fiber port and an achromatic lens. The $x$ and $y$ Galvano mirrors (Galvano scanning system, GVS002; Thorlabs) were placed in the infinity space to control the position of the stimulus. A dichroic mirror was also placed in the infinity space, and the reflected light was focused onto the sensor of a CMOS camera (Grasshopper3; Teledyne FLIR, Oregon). This design enabled monitoring of the precise location of the stimulated site[49].

An optical chopper (pulse width, 2.5 ms, MC2000B; Thorlabs) was used to activate PV interneurons with a 40 Hz pulse[72]. To investigate the suppressive effect of Dys activity, the six positions covering Dys were stimulated by a 473-nm laser focused on the brain surface by an achromatic doublet lens (AC254-60-A; Thorlabs). The centers of the stimulation sites were 430 μm apart. The power was 0.9 mW per location, and the radius was 0.5–0.75 mm, resulting in 0.51–1.15 mW/mm$^2$, with light attenuation through the skull[50]. Laser positions at both ends, position 1 (P1) and P6, were well apart from Dys, whereas P3 and P4 were centered at Dys. Given the size of the lightspot, P3 and P4 may partially stimulate neighboring paw and BF regions.

**Statistics and reproducibility.** Statistics were performed using MATLAB (R2018a, 9.4.0.813654). Data are presented as mean ± SEM, unless otherwise noted. The representative coronal sections shown in Fig. 1a, Supplementary Figs. 1b, 9b, 13a, b, and 14a, and the representative tangential sections shown in Fig. 4c and Supplementary Fig. 7 were repeated independently with the similar results at least three times or with the same number of animals shown in each experiment.

**Reporting summary.** Further information on research design is available in the Nature Research Reporting Summary linked to this article.

## Data availability
All data supporting the findings of this study are provided within the paper and its Supplementary information. A Source data file is provided with this paper. Any data are available from the authors upon request. Source data are provided with this paper.

## Code availability
All computer codes used to analyze or display the data are available from the corresponding author upon reasonable request.

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

## Acknowledgements
We thank Sachie Sekino and Yumi Tani for their excellent technical assistance. This research was supported by JSPS KAKENHI grant numbers JP15K21387, JP17H05912, JP18K14854, JP21K07285 and the Uehara Memorial Foundation to H.O. This research was also partially supported by JSPS KAKENHI grant numbers JP20H05481, JP20H05916, JP20K21508, JP17H05752, JP15H01667, and the SHISEIKAI Scholarship Fund for Basic Researcher of Medical Science, Keiko Watanabe Award to M.M. and JSPS KAKENHI grant numbers JP21K06444 to Y.U. and the program for Brain Mapping by Integrated Neurotechnologies for Disease Studies (Brain/MINDS) from the Japan Agency for Medical Research and Development, AMED, under grant number JP19dm0207057.

## Author contributions
H.O. and M.M. designed the experiments. H.O., M.K., and Y.U. performed the experiments and analyzed the data. H.O. and M.M. wrote the original draft.

## Competing interests
The authors declare no competing interests.
