## [Peer Review File · Nature Communications]

Distinct nociception processing in the dysgranular and barrel regions of the mouse somatosensory cortexREVIEWER COMMENTS

Reviewer #1 (Remarks to the Author):

In this study, the authors studied the tactile and nociceptive processing in the mouse somatosensory cortex (S1). By performing electrophysiological recording and intrinsic signal optical imaging in anesthetized mice, the authors show that whisker deflection and noxious heat is processed in distinct subregions of S1 layer 2/3: tactile input from whiskers is processed in granular barrel field (BF), while nociceptive information from whisker pad is mainly processed in dysgranular region (Dys) of S1. This spatial separation of tactile and nociceptive representation is not observed in layer 5 of S1. Optogenetic silencing Dys prevents laser heat-induced nocifensive escape behavior. In addition, the authors reported that Dys is also involved in neuropathic pain.

This is an interesting study on the role of Dys as a nociceptive processing center within S1. Overall, the experiments are well presented, data analysis is thorough, and the manuscript is clearly written. However, I do have some concerns about the findings of this work.

Major points:

1. The main finding of the study is that nociceptive and tactile information is separately processed in S1. However, the design of the experiments does not convincingly support the conclusion. In this study, the nociceptive stimulus is the noxious heat applied onto the whisker pad, while tactile input is whisker deflection. One is applied on the skin while the other is applied on the whisker. Therefore, the separation of responsive region of S1 may not have anything to do with the nature (nociceptive or tactile) of the information, but rather the location of the stimulation. To validate the finding, the authors may consider including additional noxious and tactile stimulation on the facial pad such as pinprick, noxious cold, and von Frey stimulation or innocuous heat. Does Dys always respond to noxious heat, cold and pinprick, but not to the innocuous touching/temperature of the facial pad?
2. The intrinsic signal results of neuropathic pain study (Fig. 3) are interesting, but poorly linked with the rest of paper, and the interpretation is very vague. Based on Fig. 3c and Extended Fig. 5, it is unclear whether Dys is activated in neuropathic pain. C-Fos data are helpful but it is unclear why it was measured under enriched environment. It would have been informative if the authors had performed EP recording in Dys and BF of the neuropathic pain model as they did in Figs. 1 and 2. Do Dys neurons increase activity after the nerve ligation and return to the baseline after the thread is gone? Does nerve ligation affect neuronal activity in BF?
3. Fig. 4: the authors need to verify the spatial resolution of photoactivation. In PV:: ChR2 experiments, it is unclear whether laser light only suppresses Dys but not the neighboring paw and barrel field.
4. Fig. 4: The finding that photoactivation of Po-S1 projections induces escape behavior is interesting. However, since Po projects to both Dys and BF, as shown in Extended data Fig. 10a, the results could not rule out the contribution of Po-BF projections to the escape behavior. Does photoactivation of BF have any effect on escape behavior?

Reviewer #2 (Remarks to the Author):

Review of "Distinct nociresponsive region in mouse primary somatosensory cortex" by Osaki et al.

In this manuscript the authors sought to disentangle a preferential nociceptive stimulus processing of the dysgranular cortex contained within S1 versus a preferential tactile stimulus processing of the granular barrelfield of S1 in mice. Towards this aim, the authors combined an impressive array of electrophysiological, morphological, optogenetic and behavioral assays to come to the conclusion that S1-Dys is specialized for nociception in S1. Overall, this is a very interesting subject that is of utmost relevance for basic as well as translational research. However, because most techniques are presented without adequate controls (a word that hardly exists in this manuscript) and results are inconclusive in many key aspects, I am afraid to say that the work is not in a publishable condition. In the following I will detail my major points of criticism and add some of the minor.

Mostly the terms "selective", "specific" etc. are expressed, in order to express the idea that there is some exclusiveness to noxious versus tactile responses in Dys versus BF. However, when you look at all the data (and all the recorded/analysed layers) there is always a substantial fraction of "other" responses so that the selectivity/specificity becomes a "preference of a certain degree" (as also written by the authors once: l. 147). In Fig. 2d it is easy to see that although individual neurons may be "selective", as an area, Dys has 28% of tactile "selective" cells, and in BF after all there are 14% of cells that are pain "selective". Thus, I would strongly suggest to keep phrases closer to this reality, although this reduces the story potential/impressiveness of the data. For example in l. 160/161 should read: "Furthermore, the modalities interacted with each other in a way that each suppresses the other region to a certain extent."

Talking about areas and layers: also all septa in-between barrels are considered dysgranular cortex. However, from looking at the images, I guess (but have never been told) that only the dysgranular zones surrounding the entire BF are studied and described in this manuscript. What is very strange, is the fact that all the times at which cFos is used L4 plays the major role in the analysis (and story) whereas for everything else it is completely ignored, without a reason being given. Moreover, in S1 it is an established fact that L5a is strikingly different from L5b. Unfortunately they are pooled in all electrophysiological analysis. It could be predicted that L5a in general is much more nociceptive than L5b, given the preferential projection of POM to L5a (and according to Frangeul et al., 2014).

Related to Fig. 1: This figure and related data has several problems. First, the quality of staining is low and the images are too "binary" in order to judge the specificity of the black dots, which are supposed to be specific c-Fos labeling. Because of the narrow strip of tissue, we don't see the "area specificity" of the effect. See Wagener et al. 2010 for a reasonable analysis and presentation of cFos-stainings. C-Fos expression is sensitive to all kinds of stimuli that might be more related to handling or recent cage history than to the actual experiment. What are the controls? Why is only the L4 effect significant? This is very counterintuitive when one considers the notion that specific parts of POM are responsible for relaying nociceptive stimuli - and POM projects to L5a and L1 (Keller, Jabaudon, Groh). Also automated counting on relative thick tissue (and a staining that has different intensity levels), is very problematic. How much did cross-validate your automated counting with manual analysis? Furthermore, in Fig. 1b, there is a high "baseline activity" after vehicle injection. According to Barth and Staiger labs, it is mainly L2 and L6 that has constitutive c-Fos expression and not L4. I don't see how Fig. 1f electrophysiology should be compatible with

the C-Fos staining results. I am also not sure whether this is a good way to analyze the data because there will be a lot of signal (of largely unknown identity) in what is now called noise. The better control in my opinion would be a lower temperature range, e.g. around the “indifferent temperature” (which I guess also exists for mouse).

Related to Fig. 3: ISOI is very difficult to quantify and prone to many artefacts that are nearly impossible to control. At least the example data look very much like a “draining vein artefact” which causes a lot of the enhancement in Dys at POD7 (Fig. 3b). What about constancy of illumination, level of anesthesia etc.?

Related to Fig. 4: Certainly this is a critical part for the overall attractiveness of the study. However, given the tight connection of dysgranular S1 to motor cortex (e.g. Alloway papers), could it be that (over)stimulation of e.g. POM fibers drove running as such, without a relation to the heat stimuli? In fact, if I am not mistaken, this is what Ext. Data Fig. 11 shows. So, how can you disentangle “running” from “escaping”?

Minor:

l. 124: these are very likely not “nonselective” but “non-reactive” neurons (for this type of stimulus).

p. 10, l. 278: The Ai32 reporter does not have a good reputation. How was its suitability verified?

Reviewer #3 (Remarks to the Author):

The manuscript by Osaki et al. explores the subject of involvement of the primary somatosensory cortex (S1) in mouse nociception and pain. This is a very comprehensive and insightful study, combining techniques of c-Fos labeling, intrinsic signal optical imaging, single- and multi-unit spike recording, as well as excitatory and inhibitory optogenetic modulation of cortical activity and quantifiable nocifensive behavior, all used under normal and chronic pain conditions. Noxious heat stimuli were delivered via Peltier or laser sources, which selectively activated nociceptors and avoided activation of mechanoreceptors. Anatomical tracing techniques were used to clarify the pathway from peripheral nociceptors to S1.

Together, all these techniques convincingly demonstrate that input from thermal nociceptors goes via posterior medial thalamic nucleus to the dysgranular (Dys) subdivision of S1 rather than its granular (barrel field, BF) subdivision. This input activates neurons in all cortical layers in Dys columns, whereas in the barrel-based columns it evokes a response only in layer 5. Chronic pain conditions, set up experimentally by ligation of the infraorbital nerve, shift S1 response to tactile stimulation from BF to Dys, in parallel with emerging allodynia. Selective optogenetic stimulation of nociresponsive Dys evokes nocifensive escaping behavioral responses, whereas selective optogenetic inhibition of Dys reduces behavioral responses to thermonociceptive stimulation. Overall, this study establishes the central role for Dys in nociception and neuropathic pain.

The significance of this manuscript is further enhanced by the fact that very much the same nociresponsive region has been described in the rat S1, in area 3a of nonhuman primates, and in human S1 (Panchuelo, Eldeghaidy, Marshall et al., *Neuroimage* 2020 221:117187. doi: 10.1016/j.neuroimage.2020.117187). However, the previous studies have been quite

limited in their scope, and this new investigation constitutes a major breakthrough in our appreciation of what S1 contributes to normal and pathological pain. The authors should be encouraged to emphasize the link of their findings to primates, and direct relevance to humans.

One finding of this study is surprising. As shown in Figures 1e and 2a, the response of Dys neurons to noxious heat stimuli appears to be transient – rather than continue to grow in its magnitude while the stimulus is being applied and is growing in its strength, we see an initial response when the stimulus temperature enters the noxious range, but then in less than a second the spike firing actually goes below the resting level. Such a brief, essentially an ON-response is very different from what has been reported in rat Dys and primate area 3a, where neurons exhibit slow temporal summation and reach the peak of their firing after the stimulus is terminated (similar to the time-course of slow/2nd pain).

Statistical analyses are appropriate and the experimental procedures are described in details sufficient for reproduction.

The manuscript is clearly written and there are just a few minor comments.

Specific Comments

(1) Lines 237-239: “These results indicate that Dys is mainly involved in nociception and in generating pain behaviors in S1.”

--- Results of this study do not indicate that Dys is MAINLY involved in nociception. Dys is a large region. It is well known to be involved in proprioception, and it is likely to receive inputs from other submodalities of somatosensory afferents as well (and possibly vestibular inputs). Overall, it is likely that Dys is functionally a heterogeneous region and nociception is one of its major tasks.

(2) Line 258: please change “notably” to “notable”

(3) Lines 258-267: this paragraph discusses possible pathways for Dys involvement in generating pain behavior, focusing on motor cortex and descending projections. The authors might also mention the study by Singh et al. (ref. 13), which demonstrated a major projection of S1 to anterior cingulate cortex and showed that its modulation affected pain behavior in normal and chronic pain conditions.

(4) Lines 274-275: “... we guess that the selective inhibition of nociresponsive region in S1 could ... have more therapeutic effects.”

--- This claim can be made more strongly, using Whitsel et al. 2018 review (ref. 46) as a basis.

(5) Lines 277-278: “... primate area 3a could be an evolutionally homologue to rodent Dys.”

--- It would be appropriate to cite Cooke, Padberg, Zahner, Krubitzer (Cereb. Cortex 22:1959-1978, 2012 <https://www.ncbi.nlm.nih.gov/pubmed/22021916>), since they addressed this question comprehensively.

(6) Lines 196-198 of Methods: "If the animal ran continuously on the treadmill and did not show any difference in the maximum speed between trials, the session was excluded from the analysis."

--- The rationale for this exclusion is not clear.

(7) Extended Data Fig. 4, lines 48 and 54: "panel b" should be changed to "panel a" and "panel c" should be changed to "panel b"

Oleg V. Favorov, Ph.D.
Research Associate Professor
Department of Biomedical Engineering
University of North Carolina at Chapel Hill

We thank the reviewers for their careful consideration of our paper entitled "Distinct nociresponsive region in mouse primary somatosensory cortex" (NCOMMS-21-11752-T). We appreciate their positive comments and helpful suggestions.

Below are our detailed replies to the comments.

Reviewer #1 (Remarks to the Author):

In this study, the authors studied the tactile and nociceptive processing in the mouse somatosensory cortex (S1). By performing electrophysiological recording and intrinsic signal optical imaging in anesthetized mice, the authors show that whisker deflection and noxious heat is processed in distinct subregions of S1 layer 2/3: tactile input from whiskers is processed in granular barrel field (BF), while nociceptive information from whisker pad is mainly processed in dysgranular region (Dys) of S1. This spatial separation of tactile and nociceptive representation is not observed in layer 5 of S1. Optogenetic silencing Dys prevents laser heat-induced nocifensive escape behavior. In addition, the authors reported that Dys is also involved in neuropathic pain.

This is an interesting study on the role of Dys as a nociceptive processing center within S1. Overall, the experiments are well presented, data analysis is thorough, and the manuscript is clearly written. However, I do have some concerns about the findings of this work.

Major points:

1. The main finding of the study is that nociceptive and tactile information is separately processed in S1. However, the design of the experiments does not convincingly support the conclusion. In this study, the nociceptive stimulus is the noxious heat applied onto the whisker pad, while tactile input is whisker deflection. One is applied on the skin while the other is applied on the whisker. Therefore, the separation of responsive region of S1 may not have anything to do with the nature (nociceptive or tactile) of the information, but rather the location of the stimulation. To validate the finding, the authors may consider including additional noxious and tactile stimulation on the facial pad such as pinprick, noxious cold, and von Frey stimulation or innocuous heat. Does Dys always respond to noxious heat, cold and pinprick, but not the innocuous touching/temperature of facial pad?

We thank the reviewer for the thoughtful comments. In addition to the noxious heat experiments, we conducted two new experiments, innocuous touching and noxious mechanical stimulation, on the whisker pad using the brush and von Frey filament (10 g), respectively. The results showed that Dys neurons, especially in L5b, can respond to not only noxious heat but also noxious mechanical stimulation. We describe these results in the revised manuscript (Supplementary Fig. 4, Results, Lines 146-156).

Regarding the response to innocuous stimulation in Dys, we compared neuronal responses to innocuous touching (brushing) on the whisker pad between Dys and BF. We found that the S/N ratio for the innocuous touch was much lower in Dys than in BF (Supplementary Fig. 2, Results, Lines 103-106).

Examining the temperature selectivity of Dys neurons in Fig. 3d, 55% of neurons in Dys began to respond from the innocuous temperature (33-44 °C). This observation indicates that Dys neurons responded to both innocuous and noxious temperatures. However, the distribution of the thermal threshold of Dys neurons was significantly larger in the noxious range compared with BF neurons, especially in L2/3 and L5b (Fig. 3d), suggesting that Dys neurons are well-tuned to noxious heat temperatures. Furthermore, the steepness of the peak responses to heat stimuli increased with the range of noxious heat throughout the layers of Dys (Fig. 3a). Following these results, we concluded that Dys preferentially responds to noxious stimulation rather than innocuous stimulation (Results, Lines 159-166).

2. The intrinsic signal results of neuropathic pain study (Fig. 3) are interesting, but poorly linked with the rest of paper, and the interpretation is very vague. Based on Fig. 3c and Extended Fig. 5, it is unclear whether Dys is activated in neuropathic pain. C-Fos data are helpful but it is unclear why it was measured under enriched environment. It would have been informative if the authors had performed EP recording in Dys and BF of the neuropathic pain model as they did in Figs. 1 and 2.

Do Dys neurons increase activity after the nerve ligation and return to the baseline after the thread is gone?

Does nerve ligation affect neuronal activity in BF?

We thank the reviewer for the comment. According to the suggestion, we recorded the neuronal activities from Dys and BF simultaneously at POD7 and POD21 in ION-ligated mice. Both Dys and BF activities evoked by whisker stimulation were variable and relatively lower at POD7 than expected from the intrinsic imaging. In some cases, Dys activity was somewhat higher than BF. In contrast, the neuronal activity in BF was significantly higher than Dys at POD21 in all animals.

The discrepancy between the intrinsic signal and the neural activity at POD7 may be due to differences in the measurement methods. The intrinsic signal measurement was a series of spatial summations of neuronal activities recorded from the same animals and showed relative changes of oxygen consumption between Dys and BF, whereas the electrophysiological recordings represented neuronal activities at the localized site of Dys and BF from different animals on POD7 and POD21.

Given these results, we may have underestimated the level of Dys activity. Therefore, we measured the cFos expression, a marker of neural activity, in ION-ligated mice during normal behavior (Supplementary Fig. 8) .

We apologize for the lack of explanation for why we measured c-Fos in the enriched environment condition. c-Fos is effectively expressed in SI by whisker stimulation during exploratory behavior in a novel, enriched environment (Bisler, Schleicher et al. 2002, Staiger, Masannek et al. 2002). Therefore, c-Fos expression under such conditions provides meaningful information on how Dys and BF respond during exploratory behavior under the pathophysiological pain condition.

We added the new results of the electrophysiology at POD 7 and 21 (Results, Lines 202-207, Supplementary Fig. 7) and explained the rationale for the c-Fos study under the enriched environment in the revised manuscript (Results, Lines 207-211, Supplementary Fig. 8).

3. Fig. 4: the authors need to verify the spatial resolution of photoactivation. In PV:: ChR2 experiments, it is unclear whether laser light only suppresses Dys but not the neighboring paw and barrel field.

We are sorry for the unclear explanation about how the photostimulation was localized. The power was 0.9 mW per location with a 0.5 ~ 0.75 mm radius. Because the width of Dys is approximately ~0.4 mm in the tangential view shown in Fig.4c, it is not easy to stimulate only Dys without stimulation of the neighboring paw region or BF. Therefore, we changed the positions of the laser stimulation at six points and compared the probability of changing the behavior, similar to previous studies (Guo, Li et al. 2014, Zátka-Haas, Steinmetz et al. 2021). (Fig. 5f and g). We added an explanation about this point in the Results (Lines 238-243) and Methods (Lines 255-260) of the revised manuscript, as described below.

"Since the Dys is a narrow region (approximately 0.4 mm) that is adjacent to the BF (tangential section, Fig.4c), stimulating Dys while excluding BF or the paw region is challenging. We, therefore, changed the laser stimulation positions at six points and then compared the behavior. Photoinhibition at position 3 (P3) and P4, which mainly covered ..." (Results, Lines 238-243)

"The centers of the stimulation sites were 430 μ m apart. The power was 0.9 mW per location, and the radius was 0.5–0.75 mm, resulting in 0.51–1.15 mW/mm², with the light attenuation through the skull. Laser positions at both ends, position 1 (P1) and P6, were well apart from the Dys, whereas P3 and P4 were centered at the Dys. Given the size of the

lightspot, P3 and P4 may partially stimulate neighboring paw and BF regions." (Methods, Line 255-260)

4. Fig. 4: The finding that photoactivation of Po-S1 projections induces escape behavior is interesting. However, since Po projects to both Dys and BF, as shown in Extended data Fig. 10a, the results could not rule out the contribution of Po-BF projections to the escape behavior. Does photoactivation of BF have any effect on escape behavior?

We thank the reviewer for this fundamental question. Unfortunately, we could not compare equally the effects of photoactivation between Dys and BF, because the projection patterns from POM are pretty different: The axons of POM project directly into L4 of Dys (Koralek, Jensen et al. 1988), but they are focused in L5a of BF and more weakly in L1 (Petreanu, Mao et al. 2009, Pouchelon, Gambino et al. 2014). Furthermore, the thalamocortical inputs to L1 and L5a have different impacts on neural firing (Larkum, Senn et al. 2004). Therefore, there are limitations in comparing the effects of photoactivation, which decays with the depth of brain tissue.

Because of this limitation, we also compared the effects of inactivation on escape behavior between Dys and BF. Dys inactivation (Position 3 and 4) effectively reduced the escape behavior, while BF inactivation did not (Position 5 and 6) (Fig. 5). However, these results do not entirely exclude the possibility of BF deep layers having a role in pain behavior. In the revised manuscript, we investigated many integrative type neurons at L5b in both Dys and BF (Fig. 3c). Castro et al. reported that BF activation reduced pain behavior through the feedforward inhibition of corticotrigeminal axons from L5 (Castro, Raver et al. 2017), suggesting that Dys and BF might have the opposite effect on escape or pain behavior.

We describe these points in the the Discussion of the revised manuscript (Lines 311-321). We also moved the figure for direct photoactivation of the POM-Dys pathway from the main manuscript (Figures 4h-j in the old manuscript) to the supplementary data (Supplementary Fig. 13).

References

- Bisler, S., A. Schleicher, P. Gass, J. H. Stehle, K. Zilles and J. F. Staiger (2002). "Expression of c-Fos, ICER, Krox-24 and JunB in the whisker-to-barrel pathway of rats: time course of induction upon whisker stimulation by tactile exploration of an enriched environment." *Journal of Chemical Neuroanatomy* 23(3): 187-198.
- Castro, A., C. Raver, Y. Li, O. Uddin, D. Rubin, Y. Ji, R. Masri and A. Keller (2017). "Cortical Regulation of

Nociception of the Trigeminal Nucleus Caudalis." *The Journal of Neuroscience* 37(47): 11431-11440.

Guo, Z. V., N. Li, D. Huber, E. Ophir, D. Gutnisky, J. T. Ting, G. Feng and K. Svoboda (2014). "Flow of Cortical Activity Underlying a Tactile Decision in Mice." *Neuron* 81(1): 179-194.

Koralek, K.-A., K. F. Jensen and H. P. Killackey (1988). "Evidence for two complementary patterns of thalamic input to the rat somatosensory cortex." *Brain Research* 463(2): 346-351.

Larkum, M. E., W. Senn and H.-R. Lüscher (2004). "Top-down Dendritic Input Increases the Gain of Layer 5 Pyramidal Neurons." *Cerebral Cortex* 14(10): 1059-1070.

Petreaanu, L., T. Mao, S. M. Sternson and K. Svoboda (2009). "The subcellular organization of neocortical excitatory connections." *Nature* 457(7233): 1142-1145.

Pouchelon, G., F. Gambino, C. Bellone, L. Telley, I. Vitali, C. Lüscher, A. Holtmaat and D. Jabaudon (2014). "Modality-specific thalamocortical inputs instruct the identity of postsynaptic L4 neurons." *Nature* 511(7510): 471.

Staiger, J. F., C. Masannek, S. Bisler, A. Schleicher, W. Zusratter and K. Zilles (2002). "Excitatory and inhibitory neurons express c-Fos in barrel-related columns after exploration of a novel environment." *Neuroscience* 109(4): 687-699.

Zatka-Haas, P., N. A. Steinmetz, M. Carandini and K. D. Harris (2021). "Sensory coding and the causal impact of mouse cortex in a visual decision." *eLife* 10.

Reviewer #2 (Remarks to the Author):

Review of "Distinct nociresponsive region in mouse primary somatosensory cortex" by Osaki et al.

In this manuscript the authors sought to disentangle a preferential nociceptive stimulus processing of the dysgranular cortex contained within S1 versus a preferential tactile stimulus processing of the granular barrelfield of S1 in mice. Towards this aim, the authors combined an impressive array of electrophysiological, morphological, optogenetic and behavioral assays to come to the conclusion that S1-Dys is specialized for nociception in S1. Overall, this is a very interesting subject that is of utmost relevance for basic as well as translational research. However, because most techniques are presented without adequate controls (a word that hardly exists in this manuscript) and results are inconclusive in many key aspects, I am afraid to say that the work is not in a publishable condition. In the following I will detail my major points of criticism and add some of the minor.

Mostly the terms "selective", "specific" etc. are expressed, in order to express the idea that there is some exclusiveness to noxious versus tactile responses in Dys versus BF. However, when you look at all the data (and all the recorded/analysed layers) there is always a substantial fraction of "other" responses so that the selectivity/specificity becomes a "preference of a certain degree" (as also written by the authors once: l. 147). In Fig. 2d it is easy to see that although individual neurons may be "selective", as an area, Dys has 28% of tactile "selective" cells, and in BF after all there are 14% of cells that are pain "selective". Thus, I would strongly suggest to keep phrases closer to this reality, although this reduces the story potential/impressiveness of the data. For example in l. 160/161 should read: "Furthermore, the modalities interacted with each other in a way that each suppresses the other region to a certain extent."

We thank the reviewer for the helpful comments about the terms 'selective' and 'specific'. Although Dys preferred to respond to noxious stimulations in our study, as the reviewer explains, we should not conclude that Dys is the area exclusive for nociception. Indeed, it was proposed that BF also responds to noxious inputs (Lamour, Guilbaud et al. 1983, Sun, Yang et al. 2006, Frangeul et al. 2014). Still, Dys processes more heterogeneous inputs, such as proprioception and higher-order tactile information from the P_{Om} (Chapin and Lin 1984, El-Boustani, Sermet et al. 2020). Accordingly, we have rewritten our descriptions throughout the manuscript as follows.

"Dys neurons are more responsive to noxious input, whereas BF neurons prefer tactile input."(Abstract, Lines 31-32)

"The comparisons of simultaneously recorded neural pairs showed that Dys neurons are more responsive to noxH than BF neurons" (Results, Lines 81-82)

Also, "Furthermore, the modalities interacted with each other in a way that each suppresses the other region." (Results, Line 181 in the previous manuscript) was removed from the revised manuscript because it was not correct according to the analysis of L5a and L5b.

Talking about areas and layers: also all septa in-between barrels are considered dysgranular cortex. However, from looking at the images, I guess (but have never been told) that only the dysgranular zones surrounding the entire BF are studied and described in this manuscript.

We apologize for the unclear explanation about the dysgranular zone and septa. As mentioned by the reviewer, we mainly studied the dysgranular zone in the comparison with BF. Therefore, we have added the following explanation of the dysgranular zone and septa in the revised manuscript (Results, Lines 70-72).

"The septa within the BF belong to the dysgranular region in terms of cytoarchitecture; in the following analysis, however, the dysgranular zone surrounding BF is designated as Dys."

What is very strange, is the fact that all the times at which cFos is used L4 plays the major role in the analysis (and story) whereas for everything else it is completely ignored, without a reason being given.

We thank the reviewer for pointing this out. While L4 of Dys receives direct thalamic POM input (El-Boustani, Sermet et al. 2020), L2/3 and L5 are output layers that are strongly connected with the areas involved in pain behavior or affective pain, such as motor or anterior cingulate cortices (Singh, Patel et al. 2020). To explore the neural mechanisms under the pain behavior observed in Fig. 5 of the revised manuscript (Fig. 4 in the previous manuscript), we focused on L2/3 and 5. We have added an explanation (Results, Lines 112-115), as described below.

"L2/3 and 5 are the cortical output layers that make connections with the motor and anterior cingulate cortices, which are implicated in pain behavior. Therefore, we focused on differences in nociceptive information processing in L2/3 and 5."

Moreover, in S1 it is an established fact that L5a is strikingly different from L5b. Unfortunately they are pooled in all electrophysiological analysis. It could be predicted that L5a in general is much more nociceptive than L5b, given the preferential projection of POM to L5a (and according to Frangeul et al., 2014).

Following this comment, we reanalyzed L5a and 5b separately with the probe positions and found a difference in the distribution of cell types (nociceptive, tactile, integrative) between the two. In L5a, 50% of Dys neurons showed the "nociceptive type", a preference for noxious heat, but this number was 34% for BF. In contrast, there was no difference in cell types in L5b between Dys and BF except the preference for the thermal threshold (Results, Lines 159-166, Fig. 3d). We found a large number of integrative type neurons in L5b (~50%) compared to L5a (~30%) in both areas (Results, Lines 144-145, Fig. 3c). Furthermore, we found that many neurons that responded to noxious mechanical stimulation were detected in L5b of both areas (Results, Lines 146-156, Supplementary Fig. 4). These results indicate the hierarchical processing of nociceptive information between layers in S1.

Related to Fig. 1: This figure and related data has several problems.

First, the quality of staining is low and the images are too "binary" in order to judge the specificity of the black dots, which are supposed to be specific c-Fos labeling. Because of the narrow strip of tissue, we don't see the "area specificity" of the effect. See Wagener et al. 2010 for a reasonable analysis and presentation of cFos-stainings.

We thank the reviewer for the many helpful suggestions and comments about the c-Fos images. We carefully conducted the c-Fos experiments again using a different antibody (226 003, Synaptic Systems GmbH, Göttingen, Germany) (supplementary figure 1; Fig. 1 in the previous manuscript). To ensure c-Fos expression, the animals were perfused 3 hours after capsaicin injection. We also performed co-immunostaining with VGluT2 (a marker of thalamocortical terminals), and NeuN in the same brain slices to identify the area and layer specificity. We describe these results in the revised manuscript (Results, Lines 91-97, supplementary figure 1).

C-Fos expression is sensitive to all kinds of stimuli that might be more related to handling or recent cage history than to the actual experiment. What are the controls?

In the revised manuscript, we conducted the c-Fos study for capsaicin/vehicle again. To ensure that the c-Fos expression was not affected by factors other than capsaicin/vehicle, the animals were transferred to their home cages from the animal facility when injected with capsaicin or

vehicle. Additionally, the animals were anesthetized with 2% isoflurane before and during the injection. After that, they were returned to their home cages and put back to the animal facility before recovering from the anesthesia. Thus, we carefully minimized the effects of handling and cage history on the c-Fos expression. We describe these points in the revised manuscript (Methods, lines 89-92).

Why is only the L4 effect significant? This is very counterintuitive when one considers the notion that specific parts of POM are responsible for relaying nociceptive stimuli - and POM projects to L5a and L1 (Keller, Jabaudon, Groh).

Furthermore, in Fig. 1b, there is a high "baseline activity" after vehicle injection. According to Barth and Staiger labs, it is mainly L2 and L6 that has constitutive c-Fos expression and not L4.

Because the atypical expression pattern of c-Fos in L4 may be influenced by factors other than capsaicin, such as the recovery level from anesthesia, we perfused the animals 3 hours after the capsaicin injection to ensure the stable expression of c-Fos by capsaicin in new experiments. The new results showed that the expression of c-Fos was significantly increased not only in Dys L4 but also BF L5a, which is consistent with a previous report (Frangeul, Jabaudon et al. 2014).

The new results showed the "baseline activity" after vehicle injection was relatively lower than in the previous manuscript, especially in L4 and L5a. Thus, a similar tendency as in previous studies (Bisler, Staiger et al. 2002; Frangeul, Jabaudon et al. 2014) was observed. We describe these results in the revised manuscript (Results, Lines 91-97; Supplementary Fig. 1b-c; Methods, Lines 89-92).

Also automated counting on relative thick tissue (and a staining that has different intensity levels), is very problematic.

How much did cross-validate your automated counting with manual analysis?

For the c-Fos study after the capsaicin/vehicle injection, the automatic counting was changed to manual counting due to the difference in the staining method. We have performed new co-immunostaining for c-Fos, VGluT2, and NeuN after the capsaicin injection, and the number of c-Fos expressing cells was counted manually (Results, Lines 91-97; Supplementary Fig. 1; Methods, Lines 93 - 119).

For the c-Fos study in an enriched environment, three slices were counted manually and compared with the results of the automatic counting. The manual counting was performed

by a person who was not involved in the automatic counting. The number of c-Fos positive cells in the automatic count tended to be lower than in the manual count. However, this trend was similar between regions. Therefore, changes in the expression of c-Fos were detected correctly. This explanation is described in the Methods (Methods, Lines 136-139).

I don't see how Fig. 1f electrophysiology should be compatible with the C-Fos staining results.

The expression of c-Fos was increased in BF L5a as well as in Dys L4 in the new c-Fos study after capsaicin injection (Results, Lines 91-97; Supplementary Fig. 1). Therefore, these results are compatible with the electrophysiological results in Fig. 1f.

I am also no sure whether this is a good way to analyze the data because there will be a lot of signal (of largely unknown identity) in what is now called noise. The better control in my opinion would be a lower temperature range, e.g. around the "indifferent temperature" (which I guess also exists for mouse).

We also tested a lower temperature range before the temperature increased as "noise" (N) (see figure below). As shown in **a**, the S/N values were slightly increased compared with those in Fig. 1b of the revised manuscript. But the tendencies observed in Fig. 1c and 1d were unchanged (compare **b** and **c** with Fig. 1c-d). Because we want to detect noxious heat-sensitive cells, we used the innocuous temperature range (about 33-44 °C) as "noise". We added comments about this in the Results (Lines 88-90).

Figure related to Fig. 1

a, The noise (N) range was set to a steady-state temperature before the Peltier device started heating up (gray shaded area, around 30°C). **b**, the same scatter plot shown in Fig. 1c but with the N range set to around 30°C. **c**, the same bar graph shown in Fig. 1d but the with the N range set to around 30°C.

Related to Fig. 3: ISOI is very difficult to quantify and prone to many artefacts that are nearly impossible to control. At least the example data look very much like a "draining vein artefact" which causes a lot of the enhancement in Dys at POD7 (Fig. 3b). What about constancy of illumination, level of anesthesia etc.?

We thank the reviewer for pointing this out. The luminance was constant across each IOSI. The level of anesthesia was constantly controlled by monitoring the respiratory rate (70-120 cycles/s) to reduce the contributions from draining veins as much as possible.

We also conducted ISOI signal measurements from the same animal using an absorbable surgical thread to reduce other experimental artifacts. We confirmed that the areas with a higher ISOI signal "Before", "injury (POD7)", and "Recovery (POD21)" were anatomically well matched with BF, Dys, and BF, respectively. Notably, the recovery of the signal in BF was constantly observed in all animals. Furthermore, Dys was more activated than BF in the c-Fos study (Supplementary Fig. 8), suggesting that the influence of the "draining vein artifact" is relatively minor.

Related to Fig. 4: Certainly this is a critical part for the overall attractiveness of the study. However, given the tight connection of dysgranular S1 to motor cortex (e.g. Alloway papers), could it be that (over)stimulation of e.g. POm fibers drove running as such, without a relation to the heat stimuli? In fact, if I am not mistaken, this is what Ext. Data Fig. 11 shows. So, how can you disentangle "running" from "escaping"?

We thank the reviewer for this comment.

ChR2 activation of the POm-Dys fiber during the innocuous heat stimulation did not just induce "running", because mice clearly showed a change in their particular running direction to escape from the heat stimulus (Supplementary Fig. 13d). The change of the running direction was consistent with the inhibition of Dys (Supplementary Fig. 9b). Such deliberate escape behavior suggests that neuronal networks in other pain-processing areas as well as the motor cortex are involved. Indeed, Dys sends outputs not only to the motor cortex but also to the anterior cingulate cortex, which contributes to affective pain and pain aversion (Singh, Patel et al. 2020). Furthermore, Dys connects to the secondary somatosensory cortex, which is involved in the processing of nociceptive information (Ploner, Schmitz et al. 1999, Gauriau and Bernard 2004).

Therefore, the escape behavior induced by the photoactivation of the POM-Dys fiber is the result of activation of not only the motor cortex but also other pain processing areas. We have added comments about this point in the Discussion of the revised manuscript (Lines 300-310).

Minor:

I. 124: these are very likely not "nonselective" but "non-reactive" neurons (for this type of stimulus).

As suggested by the reviewer, we have changed "nonselective" to "non-reactive" in Fig. 2 and the related Results section.

ex. "To quantify these observations, we classified the neurons into nociceptive, tactile, integrative, and non-reactive types according to S/Ns to noxH and tactile stimuli (Fig. 3b and Supplementary Fig. 3)." (Results, Line 132)

p. 10, l. 278: The Ai32 reporter does not have a good reputation. How was its suitability verified?

We used the Ai32 reporter line and verified that it can be used for cortical inhibition studies by cell attached recordings (Itokazu, Osaki et al. 2018, Sato, Osaki et al. 2019). The Ai32 reporter line has been widely used in similar cortical inactivation studies elsewhere (e.g., Guo, Li et al. 2014, Zátka-Haas, Steinmetz et al. 2021). Although we found that repetitive cortical inhibition might induce cortical plasticity (Sato et al. 2019), we carefully avoided such plasticity from affecting the results.

References

- Chapin, J. K. and C. S. Lin (1984). "Mapping the body representation in the SI cortex of anesthetized and awake rats." *Journal of Comparative Neurology* 229(2).
- El-Boustani, S., B. S. Sermet, G. Foustoukos, T. B. Oram, O. Yizhar and C. C. H. Petersen (2020). "Anatomically and functionally distinct thalamocortical inputs to primary and secondary mouse whisker somatosensory cortices." *Nature Communications* 11(1): 3342.
- Frangeul, L., C. Porrero, M. Garcia - Amado, B. Maimone, M. Maniglier, F. Clascá and D. Jabaudon (2014). "Specific activation of the paralemniscal pathway during nociception." *European Journal of Neuroscience* 39(9): 1455-1464.
- Gauriau, C. and J.-F. Bernard (2004). "Posterior Triangular Thalamic Neurons Convey Nociceptive Messages to the Secondary Somatosensory and Insular Cortices in the Rat." *The Journal of Neuroscience* 24(3): 752-761.
- Guo, Z. V., N. Li, D. Huber, E. Ophir, D. Gutnisky, J. T. Ting, G. Feng and K. Svoboda (2014). "Flow of Cortical

Activity Underlying a Tactile Decision in Mice." *Neuron* 81(1): 179-194.

Itokazu, T., M. Hasegawa, R. Kimura, H. Osaki, U.-R. R. Albrecht, K. Sohya, S. Chakrabarti, H. Itoh, T. Ito, T. K. Sato and T. R. Sato (2018). "Streamlined sensory motor communication through cortical reciprocal connectivity in a visually guided eye movement task." *Nature communications* 9(1): 338.

Lamour, Y., G. Guilbaud and J. C. Willer (1983). "Rat somatosensory (SmI) cortex: II. Laminar and columnar organization of noxious and non-noxious inputs." *Experimental Brain Research* 49(1): 46-54.

Ploner, M., F. Schmitz, H.-J. Freund and A. Schnitzler (1999). "Parallel Activation of Primary and Secondary Somatosensory Cortices in Human Pain Processing." *Journal of Neurophysiology* 81(6): 3100-3104.

Sato, T. R., T. Itokazu, H. Osaki, M. Ohtake, T. Yamamoto, K. Sohya, T. Maki and T. K. Sato (2019). "Interhemispherically dynamic representation of an eye movement-related activity in mouse frontal cortex." *eLife* 8.

Singh, A., D. Patel, A. Li, L. Hu, Q. Zhang, Y. Liu, X. Guo, E. Robinson, E. Martinez, L. Doan, B. Rudy, Z. S. Chen and J. Wang (2020). "Mapping Cortical Integration of Sensory and Affective Pain Pathways." *Current biology : CB*.

Sun, J. J., J. W. Yang and B. C. Shyu (2006). "Current source density analysis of laser heat-evoked intra-cortical field potentials in the primary somatosensory cortex of rats." *Neuroscience* 140(4): 1321-1336.

Zatka-Haas, P., N. A. Steinmetz, M. Carandini and K. D. Harris (2021). "Sensory coding and the causal impact of mouse cortex in a visual decision." *eLife* 10.

Reviewer #3 (Remarks to the Author):

The manuscript by Osaki et al. explores the subject of involvement of the primary somatosensory cortex (S1) in mouse nociception and pain. This is a very comprehensive and insightful study, combining techniques of c-Fos labeling, intrinsic signal optical imaging, single- and multi-unit spike recording, as well as excitatory and inhibitory optogenetic modulation of cortical activity and quantifiable nocifensive behavior, all used under normal and chronic pain conditions. Noxious heat stimuli were delivered via Peltier or laser sources, which selectively activated nociceptors and avoided activation of mechanoreceptors. Anatomical tracing techniques were used to clarify the pathway from peripheral nociceptors to S1.

Together, all these techniques convincingly demonstrate that input from thermal nociceptors goes via posterior medial thalamic nucleus to the dysgranular (Dys) subdivision of S1 rather than its granular (barrel field, BF) subdivision. This input activates neurons in all cortical layers in Dys columns, whereas in the barrel-based columns it evokes a response only in layer 5. Chronic pain conditions, set up experimentally by ligation of the infraorbital nerve, shift S1 response to tactile stimulation from BF to Dys, in parallel with emerging allodynia. Selective optogenetic stimulation of nociresponsive Dys evokes nocifensive escaping behavioral responses, whereas selective optogenetic inhibition of Dys reduces behavioral responses to thermonoxious stimulation. Overall, this study establishes the central role for Dys in nociception and neuropathic pain.

The significance of this manuscript is further enhanced by the fact that very much the same nociresponsive region has been described in the rat S1, in area 3a of nonhuman primates, and in human S1 (Panchuelo, Eldeghaidy, Marshall et al., *Neuroimage* 2020 221:117187. doi: 10.1016/j.neuroimage.2020.117187). However, the previous studies have been quite limited in their scope, and this new investigation constitutes a major breakthrough in our appreciation of what S1 contributes to normal and pathological pain. The authors should be encouraged to emphasize the link of their findings to primates, and direct relevance to humans.

We thank the reviewer for this positive and encouraging assessment. We added discussion about the link of our findings to primates and humans as below (Discussion, Lines 327-331).

"Neurons in non-human primate area 3a, a dysgranular region in terms of its cytoarchitecture⁵⁴, respond to noxious^{8,54,55} and proprioceptive inputs⁵⁶. These observations and phylogenetic relationships⁵⁷ suggest that primate area 3a is an evolutionary homolog to the rodent Dys. Moreover, human area 3a may play a central role in pain perception⁵⁸."

One finding of this study is surprising. As shown in Figures 1e and 2a, the response of Dys neurons to noxious heat stimuli appears to be transient – rather than continue to grow in its magnitude while the stimulus is being applied and is growing in its strength, we see an initial response when the stimulus temperature enters the noxious range, but then in less than a second the spike firing actually goes below the resting level. Such a brief, essentially an ON-response is very different from what has been reported in rat Dys and primate area 3a, where neurons exhibit slow temporal summation and reach the peak of their firing after the stimulus is terminated (similar to the time-course of slow/2nd pain). Statistical analyses are appropriate and the experimental procedures are described in details sufficient for reproduction.

The manuscript is clearly written and there are just a few minor comments.

Specific Comments

(1) Lines 237-239: "These results indicate that Dys is mainly involved in nociception and in generating pain behaviors in S1."

--- Results of this study do not indicate that Dys is MAINLY involved in nociception. Dys is a large region. It is well known to be involved in proprioception, and it is likely to receive inputs from other submodalities of somatosensory afferents as well (and possibly vestibular inputs). Overall, it is likely that Dys is functionally a heterogeneous region and nociception is one of its major tasks.

We thank the reviewer for pointing this out. We have corrected these words.

"These findings indicate that the Dys is involved in nociception and in the generation of pain behaviors." (Discussion, Lines 276-277)

(2) Line 258: please change "notably" to "notable"

We have corrected the misspelling.

"The notable finding in the present study is that Dys, but not BF, is involved in generating pain behavior (Fig.4)." (Results, Line 297).

(3) Lines 258-267: this paragraph discusses possible pathways for Dys involvement in generating pain behavior, focusing on motor cortex and descending projections. The authors might also mention the study by Singh et al. (ref. 13), which demonstrated a major projection of S1 to anterior cingulate cortex and showed that its modulation affected pain behavior in normal and chronic pain conditions.

We added the sentence, " Dys also projects to the anterior cingulate cortex²⁷, which integrates sensory and affective pain and modulates pain behavior¹³." in the Discussion of the revised manuscript (Lines 305-306).

(4) Lines 274-275: "... we guess that the selective inhibition of nociresponsive region in S1 could ... have more therapeutic effects."

--- This claim can be made more strongly, using Whitsel et al. 2018 review (ref. 46) as a basis.

Accordingly, we have modified the last paragraph of the Discussion, as described below (Lines 324-331).

"Given that manipulation of Dys activity controls pain behavior (Fig. 5), selective inhibition of the nociresponsive region in S1 may be a therapeutic target in clinical pain management. Cerebral cortical lesions in S1 causes permanent pain relief (reviewed in Whitsel et al.⁵⁴). Neurons in non-human primate area 3a, a dysgranular region in terms of its cytoarchitecture⁵⁴, respond to noxious^{8,54,55} and proprioceptive inputs⁵⁶. These observations and phylogenetic relationships⁵⁷ suggest that primate area 3a is an evolutionary homolog to the rodent Dys. Moreover, human area 3a may play a central role in pain perception⁵⁸."

(5) Lines 277-278: "... primate area 3a could be an evolutionally homologue to rodent Dys."

--- It would be appropriate to cite Cooke, Padberg, Zahner, Krubitzer (Cereb. Cortex 22:1959-1978, 2012 <https://www.ncbi.nlm.nih.gov/pubmed/22021916>), since they addressed this question comprehensively.

We thank the reviewer for introducing the paper. We have cited it in the Discussion (Line 328).

(6) Lines 196-198 of Methods: "If the animal ran continuously on the treadmill and did not show any difference in the maximum speed between trials, the session was excluded from the analysis."

--- The rationale for this exclusion is not clear.

We have added an explanation in the Methods, as described below (Methods, Lines 230 – 234).

"If the animal ran continuously on the treadmill prior to the onset of stimulus, the effect of the stimulus would be masked. Such animals tended to run continuously, regardless of the

stimuli. Therefore, no difference in the maximum speed between trials could be observed, and such sessions were excluded from the analysis."

(7) Extended Data Fig. 4, lines 48 and 54: "panel b" should be changed to "panel a" and "panel c" should be changed to "panel b"

We have corrected the legend (Supplementary Fig. 3; Extended Data Fig. 4 in the previous manuscript).

REVIEWER COMMENTS

Reviewer #1 (Remarks to the Author):

The authors have adequately addressed most of my previous comments and suggestions.

Minor point:

Figure legend of new Supplementary Fig. 4, line 63, I think it should be “Upper, Normalized MUA recorded from each probe” instead of “Upper, Normalized PSTHs of each MUA recorded from each probe”.

Reviewer #2 (Remarks to the Author):

This is the review of the revision of “Distinct nociresponsive region in mouse primary somatosensory cortex” by Osaki and colleagues. I want to start by thanking the authors for having improved the manuscript substantially when working along the lines of my previous criticism.

Nevertheless, having read it (and the comments of my fellow reviewer 1) thoroughly, it leaves me with mixed feelings. It certainly has been much improved and the data is much clearer. However, the existence of “labeled lines” that differ for two nearby spots on the snout (whiskers versus common fur/skin) very likely underlies the spatial distinctness of responses to innocuous (largely whisker) and noxious (largely thermal) stimuli in BF versus DYS, as presented here. That DYS has a propensity to react to noxious stimuli is nothing new (See Khasabov et al. 2020 for a recent one). But it also seems to code touch of common fur (or skin, as shown by Suppl.-Fig. 2 of the present manuscript) in-between the whiskers, and this even with a clear-cut topography (Takashima et al., 2005). The least I am concluding from this is that the Title certainly does not capture the major finding of the work presented here. Also the final sentence of the Abstract is wrong in the light of the Takashima paper.

However, there is another question I need to get answered before I can really make up my mind: How was it ensured that the stimulus sites represented the receptive fields (RF) of the recording sites? As one can see from Takashima et al., in the DYS there is a topography and of course the one for the whiskers is already “iconic” (Woolsey and van der Loos, 1970). So, for any given electrode location, how did you make sure that your stimuli hit the RF of these patches of cortex?

Given that these questions can be answered satisfactorily, I think the manuscript still has its merits because it starts to give us a glimpse about layer-specific functional interactions between 2 cortical areas, one with a bias toward pain processing (DYS) and another with a bias toward tactile information processing (BF). Furthermore, the behavioral part and its validation by several optogenetic approaches is also novel. Therefore, I am convinced that the manuscript needs a new twist to transmit its true message but then it could be a valuable contribution to further our understanding of pain processing.

I also agree with a fellow reviewer that the allodynia stuff (as a form of chronic pain model) feels somewhat displaced with respect to the rest of the paper. Maybe, apart from the above suggested re-writing, a new Title like “Acute and chronic pain processing by dysgranular somatosensory cortex and its interaction with the barrel cortex” will be helpful to better

integrate these different parts.

Apart from re-writing, I am not asking for more experiments. However, I would like to ask for more data being reported. For me, a crucial question for all the interactions between DYS and BF that we can infer from the data, is whether their basis is a direct transfer of stimuli from the thalamus or a secondary processing by intracortical circuitry. This could be distinguished to some extent by latency measurements in single-unit recordings, at least where the stimulus was not a “gradient”.

Other issues:

1. Language: it needs much improvement at any level. The habit of using too much “definitive article” and “plural” has to be corrected, as well as lapses of expression like in l. 33/34, which should probably read “In contrast, both modalities seem to converge on individual layer 5 neurons of each region to a different extent.” Other examples: l. 186 should be “Toward this aim, ...”. L. 201 has a typo, since the reference has to be to Suppl.-Fig. 6 (not 5). L. 216 “increased” instead of “activated”.
2. References: although suggested by a fellow reviewer, the literature list has a strong bias toward primate stuff whereas at the same time other rodent-relevant papers, like for instance Okada et al. 2021 (Sci Adv), are not to be found.

We thank the reviewers for their careful consideration of our paper NCOMMS-21-11752-B. We appreciate their positive comments and helpful suggestions.

Below are our detailed replies to the comments.

Reviewer #1 (Remarks to the Author):

The authors have adequately addressed most of my previous comments and suggestions.

Minor point:

Figure legend of new Supplementary Fig. 4, line 63, I think it should be “Upper, Normalized MUA recorded from each probe” instead of “Upper, Normalized PSTHs of each MUA recorded from each probe”.

We have corrected the legend (Supplementary Fig. 5; Supplementary Fig. 4 in the previous manuscript).

Reviewer #2 (Remarks to the Author):

This is the review of the revision of “Distinct nociresponsive region in mouse primary somatosensory cortex” by Osaki and colleagues. I want to start by thanking the authors for having improved the manuscript substantially when working along the lines of my previous criticism.

Nevertheless, having read it (and the comments of my fellow reviewer 1) thoroughly, it leaves me with mixed feelings. It certainly has been much improved and the data is much clearer. However, the existence of “labeled lines” that differ for two nearby spots on the snout (whiskers versus common fur/skin) very likely underlies the spatial distinctness of responses to innocuous (largely whisker) and noxious (largely thermal) stimuli in BF versus DYS, as presented here. That DYS has a propensity to react to noxious stimuli is nothing new (See Khasabov et al. 2020 for a recent one). But it also seems to code touch of common fur (or skin, as shown by Suppl.- Fig. 2 of the present manuscript) in-between the whiskers, and this even with a clear-cut

topography (Takashima et al., 2005). The least I am concluding from this is that the Title certainly does not capture the major finding of the work presented here. Also the final sentence of the Abstract is wrong in the light of the Takashima paper.

We thank the reviewer for this thoughtful comment. According to the reviewer's suggestion and following comments, we have changed the title of the manuscript to "Distinct pain processing in the dysgranular and barrel regions of the mouse somatosensory cortex".

We have also corrected the final sentence of the Abstract as follows. "We further demonstrate that Dys activity, but not BF activity, is critically involved in neuropathic pain and pain behavior. These findings provide new insights into the role of Dys for pain processing in S1." (Abstract, Lines 35 - 37)

However, there is another question I need to get answered before I can really make up my mind: How was it ensured that the stimulus sites represented the receptive fields (RF) of the recording sites? As one can see from Takashima et al., in the DYS there is a topography and of course the one for the whiskers is already "iconic" (Woolsey and van der Loos, 1970). So, for any given electrode location, how did you make sure that your stimuli hit the RF of these patches of cortex?

We are sorry for the unclear description about the receptive field and recording sites. The recording sites of the four-shank electrodes were determined by intrinsic signal imaging and always covered Dys and the barrel of row E. According to Takashima's paper (2005), Dys neurons adjacent to the barrels of row E have their receptive fields around row E whiskers. Therefore, the receptive fields of the recording sites matched the stimulated area. We describe these points in the Methods (lines 28-36).

Given that these questions can be answered satisfactorily, I think the manuscript still has its merits because it starts to give us a glimpse about layer-specific functional interactions between 2 cortical areas, one with a bias toward pain processing (DYS) and another with a bias toward tactile information processing (BF). Furthermore, the behavioral part and its validation by several optogenetic approaches is also novel. Therefore, I am convinced that the manuscript needs a new twist to transmit its true message but then it could be a valuable contribution to

further our understanding of pain processing. I also agree with a fellow reviewer that the allodynia stuff (as a form of chronic pain model) feels somewhat displaced with respect to the rest of the paper. Maybe, apart from the above suggested re-writing, a new Title like “Acute and chronic pain processing by dysgranular somatosensory cortex and its interaction with the barrel cortex” will be helpful to better integrate these different parts.

We thank the reviewer for these encouraging comments and suggested new title. Due to the limited number of words, we changed the title to “**Distinct pain processing in the dysgranular and barrel regions of the mouse somatosensory cortex**”.

Apart from re-writing, I am not asking for more experiments. However, I would like to ask for more data being reported. For me, a crucial question for all the interactions between DYS and BF that we can infer from the data, is whether their basis is a direct transfer of stimuli from the thalamus or a secondary processing by intracortical circuitry. This could be distinguished to some extent by latency measurements in single-unit recordings, at least where the stimulus was not a “gradient”.

We thank the reviewer for the comment. According to the suggestion, we calculated the onset-latency to tactile stimulation. For “tactile” cells, the onset-latency in Dys L2/3 tended to be larger than that in BF L2/3 and was similar to that in BF L5. This observation suggests that tactile-type neurons in Dys L2/3 receive tactile information not directly from the thalamus, but via intracortical connections from BF. In contrast, the onset latency of integrative-type neurons in L2/3 of both regions showed longer latency. We describe these results in the revised manuscript (Results, lines 145-149, and Supplementary Fig. 4).

We have already provided the data about onset latency to thermal stimuli as the distribution of thermal thresholds calculated from the onset temperature of each neuron’s response (Fig. 3d). However, onset latency to thermal stimuli is not suitable for this analysis because the stimuli have too high a gradient to distinguish the thalamic input from the secondary processing by intracortical circuitry.

Other issues:

1. Language: it needs much improvement at any level. The habit of using too much “definitive

article” and “plural” has to be corrected, as well as lapses of expression like in l. 33/34, which should probably read “In contrast, both modalities seem to converge on individual layer 5 neurons of each region to a different extent.” Other examples: l. 186 should be “Toward this aim, ...”. L. 201 has a typo, since the reference has to be to Suppl.-Fig. 6 (not 5). L. 216 “increased” instead of “activated”.

We are sorry for these typos and expressions. These points have been corrected. In addition, the revised manuscript has been reviewed for proof reading by a native English speaker.

2. References: although suggested by a fellow reviewer, the literature list has a strong bias toward primate stuff whereas at the same time other rodent-relevant papers, like for instance Okada et al. 2021 (Sci Adv), are not to be found.

We have added several rodent-relevant papers, including Okada et al. 2021 (Sci Adv), Ishikawa et al. 2018 (Pain), and Cichon et al. 2017 (Nat. Neurosci.).

REVIEWER COMMENTS

Reviewer #2 (Remarks to the Author):

I am happy to see that my comments fell on fruitful ground. The authors did a good job in rewriting some paragraphs and correcting some mistakes. It has become a nice paper.

We thank the reviewer for their careful consideration of our paper NCOMMS-21-11752-B.

REVIEWERS' COMMENTS

Reviewer #2 (Remarks to the Author):

I am happy to see that my comments fell on fruitful ground. The authors did a good job in rewriting some paragraphs and correcting some mistakes. It has become a nice paper.

We thank again the reviewer for lots of helpful and encouraging comments.